



# Real-time chemical speciation and source apportionment of organic aerosol components in Delhi, India, using extractive electrospray ionization mass spectrometry

Varun Kumar[1], Stamatios Giannoukos[1,a], Sophie L. Haslett[2], Yandong Tong[1], Atinderpal Singh[3,b], Amelie Bertrand[1], Chuan Ping Lee[1], Dongyu S. Wang[1], Deepika Bhattu[1,c], Giulia Stefenelli[1], Jay S. Dave[3], Joseph V. Puthussery[4], Lu Qi[1], Pawan Vats[5], Pragati Rai[1], Roberto Casotto[1], Rangu Satish[3], Suneeti Mishra[6], Veronika Pospisilova[1,d], Claudia Mohr[2], David M. Bell[1], Dilip Ganguly[5], Vishal Verma[4], Neeraj Rastogi[3], Urs Baltensperger[1], Sachchida N. Tripathi[6], André S. H. Prévôt[1], Jay G. Slowik[1]

[1]Laboratory of Atmospheric Chemistry, Paul Scherrer Institut, 5232 Villigen PSI, Switzerland

[2]Department of Environmental Science, Stockholm University, Sweden

[3]Geosciences Division, Physical Research Laboratory, Ahmedabad 380009, India

[4]University of Illinois at Urbana-Champaign, Urbana, Illinois 61801, United States

[5]Centre for Atmospheric Sciences, Indian Institute of Technology Delhi, New Delhi 110016, India

[6]Department of Civil Engineering and Centre for Environmental Science and Engineering, Indian Institute of Technology Kanpur, Kanpur 208016, India

[a]now at: ETH Zurich, Department of Chemistry and Applied Biosciences, 8093 Zurich, Switzerland

[b]now at: Department of Environmental Studies, University of Delhi, Delhi 110007, India

[c]now at: Department of Civil and Infrastructure Engineering, Indian Institute of Technology Jodhpur, 342037, India

[d]now at: Tofwerk AG, Uttigenstrasse 22, 3600 Thun, Switzerland

**Correspondence to:** Sachchida N. Tripathi (snt@iitk.ac.in), André S. H. Prévôt (andre.prevot@psi.ch) and Jay G. Slowik (jay.slowik@psi.ch)

**Abstract.** In recent years, the Indian capital city of Delhi has been impacted by very high levels of air pollution, especially during winters. Comprehensive knowledge of the composition and sources of the organic aerosol (OA), which constitutes a substantial fraction of total particulate mass (PM) in Delhi, is central to formulating effective public health policies. Previous source apportionment studies in Delhi identified key sources of primary OA (POA) and showed that secondary OA (SOA) played a major role, but were unable to resolve specific SOA sources. We address the latter through the first field deployment of an extractive electrospray ionization time-of-flight mass spectrometer (EESI-TOF) in Delhi, together with a high-resolution aerosol mass spectrometer (AMS). Measurements were conducted during the winter of 2018 /2019, and positive matrix factorization (PMF) was used separately on AMS and EESI-TOF datasets to apportion the sources of OA. AMS PMF analysis yielded three



primary and two secondary factors which were attributed to hydrocarbon-like OA (HOA), biomass burning OA (BBOA-1 and 2), more oxidized oxygenated OA (MO-OOA), and less oxidized oxygenated OA (LO-OOA). On average, 40 % of the total OA mass was apportioned to the secondary factors. The SOA contribution to total OA mass varied greatly between daytime (76.8 %, 10:00-16:00 local time (LT)) and nighttime (31.0 %, 21:00-04:00 local time).

The higher chemical resolution of EESI-TOF data allowed identification of individual SOA sources. The EESI-TOF PMF analysis in total yielded six factors, two of which were primary factors (primary biomass burning and cooking-related OA). The remaining four factors were predominantly of secondary origin: aromatic SOA, biogenic SOA, aged biomass burning SOA, and mixed urban SOA. Due to the uncertainties in the EESI-TOF ion sensitivities, mass concentrations of EESI-TOF SOA dominated factors were related to the total AMS

SOA (i.e., MO-OOA + LO-OOA) by multi-linear regression (MLR). Aromatic SOA was the major SOA component during the day-time, with 55.2 % contribution to total SOA mass (42.4 % contribution to total OA). Its contribution to total SOA, however, decreased to 25.4 % (7.9 % of total OA) during night-time. This factor was attributed to the oxidation of light aromatic compounds emitted mostly from traffic. Biogenic SOA accounted for 18.4 % of total SOA mass (14.2 % of total OA) during day-time and 36.1 % of total SOA mass (11.2 % of

total OA) during night-time. Aged biomass burning and mixed urban SOA accounted for 15.2 % and 11.0 % of total SOA mass ( 11.7 % and 8.5 % of total OA mass) during day-time respectively and 15.4 % and 22.9 % of total SOA mass (4.8 % and 7.1 % of total OA mass) during night-time, respectively. A simple dilution/partitioning model was applied on all EESI-TOF factors to estimate the fraction of observed day-time concentrations resulting from local photochemical production (SOA) or emissions (POA). Aromatic SOA, aged biomass burning, and

mixed urban SOA were all found to be dominated by local photochemical production, likely from the oxidation of locally emitted VOCs. In contrast, biogenic SOA was related to the oxidation of diffuse regional emissions of isoprene and monoterpenes. The findings of this study show that in Delhi, the night-time high concentrations are caused by POA emissions led by traffic and biomass burning, and the daytime OA is dominated by SOA, with aromatic SOA accounting for the largest fraction. Because aromatic SOA is possibly more toxic than biogenic

SOA and primary OA, its dominance during the day-time suggests an increased OA toxicity and health-related consequences for the general public.



## 1. Introduction

Atmospheric aerosols are suspensions of tiny solid or liquid particles in the air, ranging from a few nanometres (nm) to tens of micrometres (µm) in size. Aerosols can affect climate directly by scattering (including reflection) and absorbing solar radiation thereby altering the radiative balance of the earth-atmosphere system, and indirectly by acting as cloud condensation nuclei (CCN), thereby affecting the amount and lifetime of clouds (Forster et al., 2007). Aerosol particulate matter with an aerodynamic diameter less than or equal to 2.5 µm ($PM_{2.5}$), can easily be deposited deep into human lungs and induce oxidative stress and inflammation leading to various cardiovascular and respiratory diseases (Pope et al., 2009; Salvi, 2007; Shiraiwa et al., 2017). Aerosols can be composed of various species such as mineral dust, soluble inorganic species such as nitrate, sulfate, ammonium, and chloride, as well as organic and elemental carbon. It is estimated that organic aerosols (OA) can account for 20 % to 90 % of the total fine particulate mass (Carlton et al., 2009). OA is classified as either primary OA (POA), which is directly emitted into the atmosphere, or secondary OA (SOA), which is produced in the atmosphere by the oxidation of volatile organic compounds (VOCs) emitted from anthropogenic or natural processes, producing lower-volatility products that form new particles or condense onto the pre-existing aerosols. In many areas, SOA accounts for a significant portion of total OA mass (Jimenez et al., 2009). However, despite SOA being an important fraction of total OA and its toxicity (Daellenbach et al., 2020), our understanding of sources and formation processes of SOA in the atmosphere remains incomplete (Hallquist et al., 2009; Shrivastava et al., 2017). This limits our ability to accurately constrain SOA contributions in global climate models and regional air quality models and impedes efforts to understand SOA health effects.

Delhi, the capital city of India is a growing megapolis with a population of about 17 million and a population density of 11,320 persons per $km^2$ as per the most recent census, conducted in 2011 (Planning Department; Government of NCT of Delhi, 2021). It experiences high levels of air pollution and is amongst the most polluted cities in the world, with an annual mean $PM_{2.5}$ concentration of ~140 µg m$^{-3}$ (World Health Organization, 2018) which is much higher than Indian National Ambient Air Quality Standards (NAAQs) of 40 µg m$^{-3}$ for annual $PM_{2.5}$ concentration (Central Pollution Control Board, 2009). During the post-monsoon season, severe air pollution events are frequent, with $PM_{2.5}$ levels often reaching as high as 1000 µg m$^{-3}$ (Sembhi et al., 2020). Several recent studies have investigated the composition and sources of non-refractory (NR) OA in Delhi using highly time-resolved online measurements by an aerosol mass spectrometer (AMS) or an aerosol chemical speciation monitor (ACSM) (Bhandari et al., 2020; Gani et al., 2019; Lalchandani et al., 2021; Tobler et al., 2020). These studies were able to quantitatively resolve the most dominant POA sources i.e., traffic-related, hydrocarbon-like organic aerosol (HOA) and biomass burning organic aerosol (BBOA) (Bhandari et al., 2020; Lalchandani et al., 2021; Tobler et al., 2020). However, they were not able to assign specific sources to the oxygenated organic aerosol (OOA), due to the use of thermal volatilization (~600 ºC) in combination with harsh electron impact ionization (EI, ~70 eV) for ion generation in the AMS and ACSM, which results in significant fragmentations of analyte molecules and loss of molecular information. Generally, while AMS/ACSM datasets provide quantitative estimates of individual POA factors and total SOA contribution, they are able to describe SOA only in terms of bulk descriptors such as level of oxygenation (i.e., bulk O/C ratio).

To overcome these limitations on fragmentation and thermal decomposition, several continuous and semi-continuous instruments have been developed. For example, the chemical analysis of aerosol on-line particle inlet





coupled to a proton transfer reaction time-of-flight mass spectrometer (CHARON-PTR-MS) employs an aerodynamic particle lens for sampling and a thermal desorption unit for volatilization of aerosol constituents, which are then ionized using a proton transfer reaction (PTR) ionization scheme and analysed by a mass spectrometer. Although PTR-MS has a softer ionization scheme than EI, the energy remains high enough to yield

significant fragmentation of organic analytes (Eichler et al., 2015; Müller et al., 2017).

The Filter Inlet for Gases and AEROsols (FIGAERO) coupled to a high-resolution time-of-flight chemical ionization mass spectrometer (HR-TOF-CIMS) is a semi-continuous system where aerosol particles are first collected onto a polytetrafluoroethylene filter. The particle-laden filter is analysed periodically by passing heated ultra-high purity nitrogen gas through the filter. The resulting vapours are ionized by chemical ionization mass

spectrometry, e.g. with iodide adducts (Lopez-Hilfiker et al., 2014). Although FIGAERO uses a soft ionization technique, some compounds are still affected by thermal decomposition (Stark et al., 2017) and reactions of analytes on the filter (Kristensen et al., 2016). Moreover, it has a lower time resolution (>30min) compared to online techniques ($\leq$ 1 min).

Finally, a novel extractive electrospray ionization (EESI) interface was coupled to a portable high-resolution time-

of-flight mass spectrometer (EESI-TOF) (Lopez-Hilfiker et al., 2019). The EESI-TOF enables highly time-resolved measurements of OA chemical composition with detection limits on the order of 1-10 ng m$^{-3}$ for atmospherically relevant compounds, with negligible thermal decomposition and ionization-induced fragmentation. Recent field studies in Europe and China have demonstrated the advantage of the chemical resolution of  EESI-TOF for source apportionment of ambient OA (Qi et al., 2019; Stefenelli et al., 2019; Tong et

al., 2021).

Here, we deployed an AMS and an EESI-TOF for 2 weeks in Delhi and report comprehensive source apportionment results from the AMS and EESI-TOF datasets with a time resolution of ~10 min. We utilised the quantitative power of the AMS and the higher chemical resolution of EESI-TOF to derive quantitative estimates of individual sources of SOA and report the results from the first-ever deployment of EESI-TOF in Delhi. Having

a quantitative estimate of individual SOA contributing factors is a valuable advancement in understanding and predicting SOA health effects, its formation mechanisms, and devising effective mitigation policies.

## 2. Methodology

### 2.1 Campaign overview and sampling site

To understand and analyse the chemical composition and sources of various components of submicron PM in

Delhi, we conducted a wintertime campaign in South Delhi at the Indian Institute of Technology Delhi (IITD) campus (28.54° N, 77.19° E) from 31 December 2018 to 14 January 2019. A suite of particle and gas-phase instrumentation was deployed which included an extractive electrospray ionisation long-time-of-flight mass spectrometer (EESI-TOF) for conducting time-resolved measurements of the organic aerosol molecular ions, a high resolution aerosol mass spectrometer (HR-AMS) for measuring non refractory (NR) PM$_1$ composition. A

scanning mobility particle sizer (SMPS), consisting of a model 3080 DMA and model 3022 CPC (TSI, Inc., Shoreview, MN, USA) to measure the particle size distribution from 15.7 to 850.5 nm, an aethalometer (model AE33, Magee Scientific, Ljubljana, Slovenia) to measure the equivalent black carbon (eBC) concentration, and



an Xact 625i Ambient Metals Monitor (Cooper Environmental Services LLC, Tigard, Oregon, USA) to measure the mass of 35 different elements in $PM_{10}$ and $PM_{2.5}$, respectively (Rai et al., 2020a).

All instruments were housed in a temperature-controlled laboratory (22 °C) on the top floor of a four-story building (~12 m high) housing other laboratories and faculty offices. Aerosol sampling was performed through stainless steel tubing (6 mm I.D.; 8 mm O.D.) of ~3 m length. A $PM_{2.5}$ cyclone (BGI, Mesa Labs. Inc.) was installed at the inlet of the sampling line to remove larger particles. Here, we have used the co-located measurements by the HR-AMS and the EESI-TOF with the supporting data from other instruments as required. More details on the operation of the EESI-TOF and the HR-AMS are given in the following sections.

The sampling site i.e., IITD's location is representative of the Delhi urban area. The nearest source of local emissions is an arterial road located ~150 m away from the building along with emissions from residential use of solid fuels for cooking and heating and biomass burning in the nearby areas. A detailed description of this site's location as well as its demographic and geographic details are provided elsewhere (Lalchandani et al., 2021; Rai et al., 2020b; Singh et al., 2021; Wang et al., 2020).

## 2.2 Instrumentation

### 2.2.1 Extractive electrospray ionization time-of-flight mass spectrometer (EESI-TOF)

The use of EESI-TOF allows continuous and highly time-resolved measurements of organic aerosol composition on a near molecular level (i.e., chemical formulae of molecular ions) with negligible thermal decomposition or ionization induced fragmentation. A detailed description of the EESI-TOF and its operating principles was provided elsewhere (Lopez-Hilfiker et al., 2019). Briefly, the sampled aerosols first pass through a multi-channel charcoal denuder to strip gaseous components. Breakthrough of gas-phase species can cause high background signals, as previously observed in Beijing (Tong et al., 2021). In that study, a small denuder (diameter of 4 mm and length of 30 to 40 mm ) was used, whereas in the present study we used a larger denuder (i.e., with 69 channels, outer diameter: 8.5 mm, length: 60 mm). The denuder was exchanged every 48 h and regenerated by baking at ~200 °C for 12 hours. The denuder was positioned ~20 cm upstream of the electrospray and mass spectrometer inlet to avoid decreased transmission of larger particles (Tong et al., 2021). After passing through the denuder, the sampled particles intersect with a plume of electrospray (ES) droplets generated by a commercial electrospray probe and delivered through a fused silica capillary with precut tips with an inner diameter of 50 μm (BGB Analytik, AG). The ES solution used in this study consisted of a 1:1 water/acetonitrile (v/v) mixture doped with 100 ppm NaI. The ES solution was charged by applying a high potential difference (~2.5-2.7 kV) at the ES capillary tip. The flow of ES solution through the silica capillary is controlled by a high-accuracy microfluidic flow controller (Fluigent, GmbH). Upon intersection with ES droplets, the soluble fraction of analyte aerosol is extracted into the droplets. The analyte-laden droplets then pass through a heated stainless-steel capillary (~250 °C), wherein the electrospray solvent evaporates, and analyte ions are generated. Due to the short residence time (~1 ms) in the capillary, heat transfer to the particles is limited, and negligible thermal decomposition is observed. Finally, the ions are analysed by a high-resolution long-time-of-flight mass spectrometer (LTOF-MS, Tofwerk AG, Switzerland) configured for positive ion detection. In the configuration of the mass spectrometer and ionization scheme used in this study, one can detect a wide range of molecules present in the organic aerosols, including sugars, alcohols, acids, and organo-nitrates. The detected molecular classes include nearly all the compounds present in the secondary organic aerosol (SOA), with notable exceptions of organosulfates which are





typically detected as negative ions and non-oxygenated species such as alkanes and alkenes. The ions were observed predominantly as adducts with $Na^+$, i.e. $[M]Na^+$, but for simplicity are denoted herein using the neutral formula (M). For example, an ion observed as $[C_6H_{10}O_5]Na^+$ is reported as $C_6H_{10}O_5$.

The EESI-TOF sampled continuously at a flow rate of ~1 L min$^{-1}$, alternating between direct ambient sampling (5 min) and sampling through a particle filter (3 min) for measurement of the instrument background. This measurement/filter cycle is shorter than those used in previous ambient measurements to avoid clogging of the EESI capillary and to increase the spray stability. The EESI-TOF data analysis was performed using Tofware version 2.5.7 (Tofwerk AG, Switzerland). The original data was acquired at a time resolution of 1 Hz and high-resolution (HR) peak fitting was applied to data averaged to 10 seconds for the mass-to-charge (*m/z*) ratio range 145-350. In total, 1030 ion formulae were fitted. The ambient aerosol composition ($M_{diff}$) was calculated by subtracting background spectra obtained during particle filter sampling ($M_{filter}$) from the mass spectra obtained during direct ambient sampling ($M_{total}$). The background spectra ($M_{filter}$) were calculated by interpolating the average between adjacent background spectra (i.e., particle filter measurements before and after an ambient sampling period). This methodology is similar to the one reported in previous EESI-TOF studies (Qi et al., 2019; Stefenelli et al., 2019; Tong et al., 2020). After obtaining the ambient aerosol composition, the ambient aerosol spectra ($M_{diff}$) were averaged to 10 minutes for further processing. The error matrix ($\sigma_{diff}$) corresponding to $M_{diff}$ values was calculated from Poission ion counting statistics (Allan et al., 2003) from ambient sampling $\sigma_{total}(i,j)$ and filter sampling periods $\sigma_{filter}(i,j)$, added in quadrature as follows:

$$\sigma_{diff}(i,j) = \sqrt{\sigma_{total}^2(i,j) + \sigma_{filter}^2(i,j)} \qquad (1)$$

As a final step, data were filtered to remove ions whose total signal and/or variability were either dominated by the background or too noisy for meaningful interpretation. Ions that met both of the following criteria were accepted for further analysis:

1.  Ratio of signal to uncertainty i.e., $M_{diff}/\sigma_{diff}$, where $\sigma_{diff}$ represents the precision-based uncertainties as calculated by using Equation 1. Ions with a median $M_{diff}/\sigma_{diff} < 0.2$ were removed from further analysis (Paatero and Hopke, 2003).

2.  Ratio of signal to background i.e., $M_{diff}/M_{filter}$. This identifies ions whose time series is dominated by instabilities in the spray and/or background drifts due to adsorption/desorption of semi-volatile compounds. Ions with median ratio of $M_{diff}/M_{filter} < 0.1$ were removed.

In the end, 641 ions between *m/z* ranges of 150-350 were retained for further analysis.

### 2.2.2 High resolution time-of-flight aerosol mass spectrometer (HR-AMS)

The HR-AMS (Aerodyne Research Inc., Billerica, MA, USA) was equipped with a PM$_1$ aerodynamic lens and measured the composition of NR-PM$_1$. A detailed description is given elsewhere (DeCarlo et al., 2006; Canagaratna et al., 2007). Briefly, ambient air is sampled continuously through a critical orifice into a PM$_1$ aerodynamic lens, which focuses the particles into a narrow beam and accelerates them to a velocity that is inversely related to their vacuum aerodynamic diameter (Williams et al., 2013). The particles then impact a resistively heated surface (~600 °C) and flash-vaporize. The resulting gas is ionized by electron ionization (EI,





70 eV) and detected by a time-of-flight mass spectrometer. The detected ion rate measured at a specific mass to charge ratio ($m/z$) is then converted to mass concentration in µg m$^{-3}$ (Jimenez et al., 2003).

In this study, two ionization efficiency (IE) calibrations of the instrument were performed (one before the commencement of the campaign and the other at the end) using 300 nm $NH_4NO_3$ particles. More details on the instrument operation during this campaign have been reported elsewhere (Singh et al., 2021; Singh et al., 2019). The instrument was operated in V mode at a time resolution of 2 minutes. Every 30 seconds, it switched between mass spectra (MS) and particle time-of-flight (PToF) mode, completing two cycles within each integration period. The mass spectra of an ensemble of particles are measured in MS mode, whereas in the PToF mode, the particle beam is modulated by a chopper spinning at 130 Hz, resulting in the size-resolved mass spectra. Unit mass resolution (UMR) data were analysed using the SQUIRREL data analysis toolkit (version 1.59) programmed in the IGOR Pro 6.37 software environment (Wavemetrics, Inc., Portland, OR, USA). High-resolution peak fitting analysis was conducted using PIKA (version 1.19) (DeCarlo et al., 2006) for $m/z$ 12 to 120. Collection efficiency was estimated using the composition-dependent algorithm of (Middlebrook et al., 2012) implemented in SQUIRREL. Pieber correction was applied according to the method recommended by Pieber et al. (2016)

## 2.3 Source apportionment

### 2.3.1 Positive matrix factorization

Source apportionment was performed separately on the AMS OA and EESI-TOF datasets using the positive matrix factorization algorithm (PMF, Paatero and Tapper, 1994) implemented within the multilinear engine (ME-2, Paatero, 1999). In this study, the Source Finder (SoFi Pro 6.8, Datalystica Ltd.) interface was used for model configuration and post-analysis (Canonaco et al., 2013). PMF is a bilinear model that represents the sample matrix **X,** having dimensions of $m \times n$ representing $m$ measurements of $n$ variables as a product of two matrices **G** (dimensions of $m \times p$) and **F** (dimensions of $p \times n$). The number of columns in the modelled matrix **G** and rows in the modelled matrix **F** are equal to the number of factors $p$, i.e. individual sources chosen to describe the dataset. The PMF model operates under non-negativity constraints, i.e. negative values are not permitted in **G** or **F**. The PMF model is expressed as:

$$\mathbf{X} = \mathbf{G} \times \mathbf{F} + \mathbf{E} \tag{2}$$

Here **G** represents the time-dependent factor concentrations (i.e., time-series) and **F** represents the chemical composition (i.e., mass-spectrum) of the resolved factors. Model residuals are contained in **E**.

The PMF model solves Eq. 2 using a least-squares algorithm that iteratively minimizes the objective function $Q$, defined as

$$Q = \sum_{i=1}^{n} \sum_{j=1}^{m} \left( \frac{e_{ij}}{\sigma_{ij}} \right)^2 \tag{3}$$

In Eq. 3, $e_{ij}$ represents elements of the residual matrix and $\sigma_{ij}$ represents the measurement uncertainties corresponding to the input point $x_{ij}$, where $i$ and $j$ are the indices representing measurement time and variable (or integer $m/z$), respectively. The theoretical value of $Q$, denoted $Q_{\text{exp}}$, can be estimated as:

$$Q_{\text{exp}} = mn - p(m + n) \tag{4}$$



PMF is subject to rotational ambiguity, meaning that different combinations of **G** and **F** matrices exist that can yield the same or similar $Q$ values. Some of these combinations may represent environmentally unreasonable representations of the dataset. To direct the model towards interpretable rotations, *a priori* information can be introduced by constraining selected factor time-series or mass-spectra using an *a*-value approach (Canonaco et al., 2013; Crippa et al., 2014). In this method, one or more factor profiles and/or time series are constrained to resemble reference profiles or/and time series, with the scalar $a$ ($0 \leq a \leq 1$) determining the tightness of constraint. As an example, if constraints are applied to mass-spectra, the *a*-value determines the extent to which a factor mass-spectrum in the final solution ($f_{j,\text{solution}}$) is allowed to deviate from the anchor mass-spectrum ($f_j$) provided to the model as the initial starting point.

$$f_{j,\text{solution}} = f_j \pm a \cdot f_j \tag{5}$$

As an example, if an *a*-value of 0.1 is used, all the variables in the resulting mass spectrum can vary between ±10 % of the input constraining mass spectrum. Note that post-PMF normalization of factor profiles may cause the final values to slightly exceed the limits defined by Eq. 5.

### 2.3.2 Source apportionment of AMS dataset

The AMS OA matrix $\mathbf{X}_{\text{AMS}}$ consisted of organic ion time series derived from high-resolution (HR) peak fitting for *m/z* 12 to 120 and the integrated signal across integer *m/z* (unit mass resolution or UMR) for *m/z* 121 to 300. A total of 507 variables were used in sample matrix $\mathbf{X}_{\text{AMS}}$, 332 of which had chemical formulae assigned to them through HR fitting. The remaining 175 variables were UMR species. For the UMR data, we excluded *m/z* 149 due to interference from phthalic acid emitted by the servo housing, as well as *m/z* 183, 184, and 186 due to the interference from the tungsten filaments. Uncertainties were calculated according to the method by Allan et al. (2003), which accounts for counting statistics of the individual ions as well as the uncertainty in the detector response to individual ions. Variables with a signal-to-noise ratio (SNR) < 0.2 were down-weighted by a factor of 10 whereas those with SNR < 2 were down-weighted by a factor of 2 (Paatero and Hopke, 2003). Further, ions calculated from the $CO_2^+$ signal (i.e., $O^+$, $OH^+$, $H_2O^+$, and $CO^+$) were removed from $\mathbf{X}_{\text{AMS}}$ prior to PMF analysis to avoid overweighting $CO_2^+$ intensity (Ulbrich et al., 2009) and were recalculated from $CO_2^+$ during post-analysis.

As a first step, we ran the PMF in unconstrained mode with the number of factors ranging from 3 to 8. Each solution was inspected based on its $Q/Q_{\text{exp}}$ value and physical interpretation of individual factors. Large decreases in $Q/Q_{\text{exp}}$ values were observed when the number of factors increased from 3 to 5, while small incremental changes were observed when the number of factors increased beyond 5. Further, solutions with more than 5 factors yielded only additional biomass burning-related factors, the differences between which could not be physically interpreted. Hence we chose a 5-factor solution as the best representation of the data. The unconstrained PMF resulted in an HOA factor with a high degree of oxygenation i.e., O:C ~0.15 which is a factor ~3 higher than the HOA factor obtained at the same site in a recent study (Lalchandani et al., 2021). To get a cleaner HOA profile, we took the HOA factor profile from an unconstrained 8-factor solution and used it to constrain the HOA factor in the final 5-factor solution. We explored the PMF solutions with higher number of factors, but the O:C ratio of the HOA profile did not show a significant decrease for solutions with more than 8 factors.



The factors obtained from the AMS source apportionment were identified based on their correlations with external measurements, mass spectral features, diurnal trends, and relationship to anthropogenic activities as well as meteorological and environmental conditions (e.g., temperature, expected trends in human activities). The interpretation of the final 5-factor solution is discussed in Sect. 3.1.

### 2.3.3 Source apportionment of EESI-TOF dataset

A total of 641 ion formulae from $m/z$ 140-350 were used in the final PMF input matrix $\mathbf{X}_{EESI}$ of the EESI-TOF data. The initial PMF model was run without constraints for 6 to 15 factors and each solution was checked for the interpretability of the results. The 6-factor solution yielded a factor identified as primary biomass burning (characterized by ~90 % of factor profile signal from $C_6H_{10}O_5$, which is likely dominated by levoglucosan, a biomass burning tracer) and five other factors related to primary cooking emissions, aged biomass burning, and 3 SOA factors (described in Sect. 3.2). Although the main spectral and temporal features of these factors were not consistent with primary biomass burning, they nonetheless contained significant signals from $C_6H_{10}O_5$ (comprising 10 % - 15 % of the factor profiles), consistent with mathematical mixing of biomass burning into these factors. Increasing the number of factors from 6 to 10 decreased the contribution of $C_6H_{10}O_5$ in the aged biomass burning factor to ~12 %, consistent with similar factor observed in previous studies (Qi et al., 2019; Tong et al., 2021). For the non-biomass burning factors, the contribution of $C_6H_{10}O_5$ to the factor profiles decreased to < 2.5 %, while key spectral and temporal features were retained. As the number of factors increased, the newly added factor profiles all had high (>20%) contributions from $C_6H_{10}O_5$, which is characteristic of primary biomass burning. However, these new factors could not be physically interpreted, and were therefore considered to result from mathematical splitting. Increasing the number of factors to 11-15 yielded only further splitting of the primary biomass burning profiles, but no longer affected the $C_6H_{10}O_5$ contributions to the non-primary biomass burning factors.

This preliminary analysis suggested that the variability in the dataset is optimally represented by 6 factors. However, because the unconstrained 6-factor solution did not provide unmixed factors (as described above), we constructed an unmixed 6-factor solution by constraining profiles for primary cooking, aged biomass burning, and the 3 SOA factors. The reference profiles for these 5 factors were taken from the unconstrained 10-factor solution. The remaining five profiles (from the unconstrained 10-factor solution) were combined on a mass-weighted basis to form a single primary biomass burning profile. This 6-factor solution is referred to as the "base case" hereafter.

The statistical stability and uncertainties of the base case were accessed by a combined bootstrap analysis/randomized $a$-value selection (i.e., sensitivity test of the tightness of constraint). Bootstrapping was implemented by random resampling of the rows of the original data matrix and corresponding entries of the error matrix, such that in each bootstrap iteration some rows were sampled multiple times while others were not sampled at all, thus creating new matrices in each iteration of the bootstrap analysis that were of the same dimensions as the original input matrices (Davison and Hinkley, 1997; Paatero et al., 2014). Simultaneously the a-values of the 5 constrained factors (primary cooking-related, aged biomass burning, and 3 unique SOA factors) were randomly selected from within predefined limits chosen to maximize exploration of the solution space while maintaining computational efficiency. The bootstrap/$a$-value analysis was conducted in two stages: (1) an exploratory analysis on a small number of runs that was used to determine the $a$-value limits; and (2) the final analysis on 1000 bootstrap runs with $a$-value randomization occurring within these limits.





The $a$-value limits for the combined bootstrap/$a$-value randomization analysis were selected after an exploratory analysis of 250 bootstrap runs in which the $a$-values of every constrained factor were allowed to vary over the full range (0 to 1), with a step size of 0.1. The 250 individual solutions were analysed and classified as "good" or "mixed" following the method of (Stefenelli et al., 2019), which consists of the following steps: (1) calculation of the Spearman correlation coefficients between the time-series of each factor from the base case and a bootstrap solution, yielding a correlation matrix for each bootstrap run with the correlation values between bootstrap factors and corresponding base case factors on the matrix diagonal; (2) requirement that the correlation coefficient on the matrix diagonal was higher than those on the intersecting row and column by a statistically significant margin (based on a preselected significance level $p$ from a $t$-test). Solutions satisfying this requirement were classified as "good" solutions, whereas those failing this test were classified as "mixed" solutions. From visual analysis of ~50 randomly selected solutions, we selected $p = 0.3$ as the appropriate confidence level. We then assessed the acceptance probability as a function of $a$-value, selecting the $a$-value upper boundary to be the value above which 75 % of solutions were classified as mixed. The ranges of $a$-values selected for cooking-related OA and four SOA factors are given in supplementary table ST1.

The $a$-value limits obtained above were utilized in a final combined bootstrap/$a$-value randomization analysis, consisting of 1000 runs. Solutions resulting from this 1000 run bootstrap were separated into "good" and "mixed" solutions using the same acceptance/rejection criteria as used in the exploratory bootstrap. The final bootstrap analysis resulted in 835 "good" solutions out of 1000 which were kept for further analysis. The solution presented in Sect. 3 is the average of these 835 solutions.

**2.4 Estimation of the fraction attributable to local production or emissions during day-time**

In order to isolate the effects of boundary layer dynamics and gas-particle partitioning from those of photochemical production, we modelled the average concentration of all EESI-TOF factors during day-time (averaged between 10:00 to 16:00 LT) (denoted $C_{model}$) based on the average concentration of the previous night (averaged between 21:00 to 04:00 LT), assuming that all changes were driven by partitioning and/or boundary layer expansion. The dilution and partitioning effects on the SOA factors were calculated by attributing each factor to a distinct organic species with bulk properties as given in the Supplementary Table ST2. The relative difference between measured ($C_{measured}$) and modelled average day-time concentrations ($C_{model}$) is attributed to local photochemical production for SOA factors and local emissions for POA factors. This analysis was applied to each factor on a day-by-day basis.

The modelled day-time concentration of a particular factor on a day $i$, $C_{model,i}$ was calculated by combining the effects of both dilution and partitioning on the average night-time concentration, $C_{nighttime,i-1}$ of that factor observed during the previous night (i.e., day $i$-1):

$$C_{model,i} = C_{nighttime,i-1} \times D_{f,i} \times P_{f,i} \tag{6}$$

where $D_f$ is the dilution factor, i.e. the fractional change in night-time concentrations due to dilution; $P_f$ is the partitioning factor, i.e. the fractional change in the night-time concentrations due to gas-particle partitioning.

The dilution factor for each day was calculated using the ratios of planetary boundary layer heights (PBLH) during night-time and day-time. PBLH heights data was obtained from the Real-time Environmental Applications and Display sYstem (READY, (Rolph et al., 2017)) website. The PBLH data was available at a 3-hourly resolution



hence single values of PBLH obtained at 00:00 during night-time and 12:00 during day-time were used for each day.

$$D_{f,i} = PBLH_{night,i}/PBLH_{day,i}$$

The partitioning coefficient $\xi_p$ for each factor $p$ was calculated using basic partitioning theory:

$$\xi_p = \left(1 + \frac{c^*}{c_{OA}}\right)^{-1} \tag{7}$$

where $C_{OA}$ is the mass concentration of organic aerosols (measured OA mass by AMS in this study) and $c^*$ is the effective saturation vapour concentration of each factor, assuming the activity coefficient is 1 (Donahue et al., 2006). The saturation vapour concentration at room temperature, $c^*(298\ K)$ was estimated using the molecular corridor approach (Li et al., 2016), based on the framework developed originally for the two-dimensional volatility

10 basis set (Donahue et al., 2011),

$$\log_{10} c^*(298K) = (n_C^0 - n_c)b_c - n_O b_O - 2\frac{n_C n_O}{n_c + n_O} b_{CO} \tag{8}$$

where $n_C^0$ is the reference carbon number; $n_C$ and $n_O$ are the number of carbon and hydrogen atoms, respectively which are given in supplementary Table ST2; $b_C$ and $b_O$ are the corresponding parameterization values for each class of compounds (i.e. CH and CHO); $b_{CO}$ is the coefficient of carbon-oxygen non-ideality, $n_c n_O/(n_c + n_O)$,

15 hereafter referred to as $NI_{CO}$. $n_C^0$, $b_C$, $b_O$, and $b_{CO}$ values, used were 25, 0.475, 0.2, and 0.9 respectively (Mohr et al., 2019; Tröstl et al., 2016; Pankow and Asher 2008). The temperature-dependent effective saturation concentration $c^*(T)$ was calculated using the Clausius-Clapeyron equation (Li et al., 2016; Donahue et al., 2006). $\xi_p$ was calculated for each factor on an hourly basis and was later averaged to obtain single day-time and night-time values for each day. The partitioning factor $P_f$ was calculated by using Eq. 9:

$$P_{f,i} = \xi_{day,i}/\xi_{nightime,i} \tag{9}$$

Based on modeled day-time concentration ($C_{model}$) and observed day-time concentration ($C_{measured}$), the fraction of day-time concentrations attributed to local photochemical production for SOA factors or to direct emissions for POA factors was calculated using the following equation:

Fraction attributable to local production or emissions $= (C_{measured} - C_{model})/C_{measured}$ (10)

## 3. Results and discussions

### 3.1 AMS source apportionment results

From the AMS source apportionment, we identified three primary factors, namely hydrocarbon-like OA (HOA), biomass burning OA (BBOA-1 and BBOA-2), and two secondary factors, denoted more-oxidized oxygenated OA

30 (MO-OOA) and less-oxidized oxygenated OA (LO-OOA). Fig. 1 shows the factor mass spectra (Fig. 1a), time series (Fig. 1b), and diurnal trends (Fig. 1c) of all factors. The relative contributions of these factors to total OA mass on a 24-h basis, during daytime, and during night-time are shown by means of pie charts in Fig. 1d.

The HOA mass spectrum contains prominent contributions from $C_xH_y$ fragments (e.g. $C_3H_5^+$, $C_3H_7^+$, $C_4H_7^+$, $C_3H_9^+$). This is consistent with saturated and unsaturated hydrocarbons, which are major constituents of fossil



fuels. Similar factors have been observed in many previous studies and are typically associated with traffic emissions (Lanz et al., 2007; Zhang et al., 2011). The diurnal pattern (Fig. 1c) shows a small peak during the morning rush hour (07:00 LT) and a larger one during the evening rush hour (18:00-22:00 LT). The morning peak is partially obscured by the decreasing concentrations due to dilution caused by a rising boundary layer. As a

result, the HOA factor reaches its minimum during midday hours (12:00-16:00 LT). Such strong boundary layer cycling is a known characteristic of Delhi and affects nearly all primary species (Gani et al., 2019; Lalchandani et al., 2021; Tobler et al., 2020). The low temperatures during the evening hours reduce the boundary layer height, resulting in an accumulation of species. The HOA time series is well correlated with $NO_x$, further supporting the assignment of this factor to traffic-related sources, as shown in Fig. 1b.

The mass spectra of BBOA-1 and BBOA-2 both have strong signals from $C_2H_4O_2^+$ ($m/z$ 60) and $C_3H_5O_2^+$ ($m/z$ 73)fragments, which are characteristic of anyhdrosugars like levoglucosan, a product of cellulose pyrolysis (Simoneit et al., 1999). The high abundances of these fragments in BBOA mass spectra have been reported in earlier studies (Crippa et al., 2013; Zhang et al., 2011). BBOA-1 has about 1.5 % and 0.8 % of its total signal attributed to $C_2H_4O_2^+$ and $C_3H_5O_2^+$ respectively, compared to 3.9 % and 1.7 % respectively for BBOA-2. All other

factors have lower contributions from these fragments; HOA, MO-OOA, and LO-OOA have 0.3 %, 0.2 %, and 0.8 %, respectively of their total signal, respectively, attributed to $C_2H_4O_2^+$, and 0.2 %, 0.1 % and 0.4 %, respectively of their total signal, respectively, attributed to $C_3H_5O_2^+$. BBOA-1 and BBOA-2 explain 21.6 % and 36.8 % of the temporal variability of $C_2H_4O_2^+$, while LO-OOA, MO-OOA, and HOA respectively explain 20.3 %, 11.6 % and 3.1 % of its temporal variability. Similarly, for $C_3H_5O_2^+$, BBOA-1 and BBOA-2 explain 20.9 %

and 31.6 % of its temporal variability respectively, while LO-OOA, MO-OOA, and HOA explain 20.2 %, 17.8 % and 4.3 % of its temporal variability, respectively. The rest is unexplained variability. The BBOA factors also have higher contributions (relative to other factors) from high $m/z$ species e.g., 116, 118, 202, etc. which were previously associated with polycyclic aromatic hydrocarbons (PAHs) (Bruns et al., 2015; Dzepina et al., 2007).

The bulk O:C, H:C, and N:C ratios of BBOA-1 are 0.37, 1.8, and 0.05, respectively, compared to 0.47, 1.84, and

0.019 for BBOA-2. The N:C value is almost 2.5 times higher for BBOA-1 compared to BBOA-2. A nitrogen-rich solid-fuel combustion factor was identified in a previous study at the same site and was attributed to biomass combustion with possible mixing of coal and other solid fuels (Lalchandani et al., 2021). Both BBOA-1 and BBOA-2 show similar diurnal trends with an evening time increase, indicating increased emissions as well as the reduction in boundary layer height due to decreasing temperature (Gani et al., 2019; Lalchandani et al., 2021),

and very low values during day-time hours. A steep decline during mid-day hours of both BBOA-1 and BBOA-2 is attributed to less intense source contributions and an increase in boundary layer height as well the increased volatilization of semi-volatile components. A contrasting feature in diurnal trends of BBOA-1 and BBOA-2, however, is the extent to which both these factors increase in evening hours as compared to their average day-time values. While BBOA-1 increases by a factor of ~2-3 during evening hours, BBOA-2 increases by a factor of

~10 during the same time. The differences in bulk elemental ratios and diurnal patterns of the BBOA factors support their treatment as separate factors.

The primary biomass burning and aged biomass burning factors from the EESI-TOF (see sect. 3.2) also show a similar trend, with the diurnal pattern of the primary biomass burning factor from the EESI-TOF showing a factor



of ~50 increase during evening rush hours, whereas the oxidized biomass burning factor only exhibits a ~2-3 fold increase during the same time (Fig. S1).

From the two retrieved SOA factors, MO-OOA is more oxygenated with a bulk O:C ratio of 0.99, which is highest among all the factors (> 2 times higher than that of LO-OOA and BBOAs and ~12 times higher than HOA). The

mass spectrum of MO-OOA contains large contributions from $CO_2^+$ ($m/z$ = 44), consistent with OOA factors described in other studies (Ng et al., 2010). The $CO_2^+$ fragment usually arises from carboxylic acid groups in diacids or multifunctional acidic compounds (Duplissy et al., 2011). The high degree of oxygenation suggests its secondary origin (Jimenez et al., 2009; Zhang et al., 2011). Despite the aforementioned boundary layer effects, the diurnal trend of MO-OOA shows an increase during the day, implying formation occurs as a result of day-

time photochemical reactions, although the sources/precursors cannot be inferred from the AMS factor spectrum. Overall, MO-OOA correlates well with $SO_4$ measured by the AMS.

LO-OOA contains a lower contribution from $CO_2^+$ (though still higher than any of the POA factors) and higher contributions from less oxygenated species. The bulk O:C ratio of this factor is 0.46. Due to the lower oxygenation and presumably higher volatility of LO-OOA, its partitioning behaviour between the gas and particle phase is

more sensitive to the ambient temperature and the total OA concentration than MO-OOA. As a result, LO-OOA exhibits increased concentrations at night (lower temperature, higher total OA).

The PMF analyses on the AMS dataset as discussed above show the relative importance of primary and secondary sources (Fig. 1d) with traffic and primary biomass burning as major contributors to the primary organic aerosol. However, while the AMS can quantify total SOA and delineate it by the extent of oxygenation and/or volatility,

it does not provide source-specific information. In the next section, we report the source apportionment results from EESI-TOF data and investigate the individual sources that could contribute to the SOA factors.

### 3.2 EESI source apportionment results

The EESI-TOF source apportionment results yielded six factors. Of these, biomass burning and cooking-related

OA were attributed to primary aerosol. The remaining four factors were attributed to secondary sources, and denoted aromatic SOA, biogenic SOA, aged biomass burning, and mixed urban SOA. These EESI-TOF factors can be qualitatively related to the AMS, with EESI-TOF primary biomass burning corresponding to AMS BBOA, and the 4 EESI-TOF secondary factors providing a more source-specific representation of AMS OOA (i.e. LV-OOA + SV-OOA). Note that cooking-related OA was retrieved only by the EESI-TOF and not the AMS (see

discussion below), while as expected the EESI-TOF did not retrieve HOA due to its insensitivity to alkanes and alkenes (see Sect. 2.2.1). Related AMS and EESI-TOF factors are compared below as appropriate.

### 3.2.1 Primary factors

**Primary biomass burning:** The mass spectrum of primary biomass burning is dominated by $C_6H_{10}O_5$, likely associated with levoglucosan and its isomers, which constitutes 81.1 % of the total mass spectral signal in this

factor (Fig. 2a). The second highest contribution to the mass spectrum comes from the ion $C_8H_{12}O_6$ (2.1 % of the total signal), which could possibly be a derivative of syringol, a prominent compound found in wood-burning smoke (Yee et al., 2013). These features are qualitatively similar to primary biomass burning mass spectra observed by EESI-TOF in the previous studies (Qi et al., 2019; Stefenelli et al., 2019; Tong et al., 2020). The next





three highest contributing ions with 0.47 %, 0.44 %, and 0.33 % of total signal, are $C_{11}H_{14}O_4$, $C_6H_{12}O_5$, and $C_6H_{10}O_4$ respectively. $C_{11}H_{14}O_4$ could be tentatively assigned to syringyl ethanone, whereas $C_6H_{10}O_4$ may be associated with methyl-glutaric acid. Both of these compounds have previously been found in biomass burning smoke (Bertrand et al., 2018; Qi et al., 2019).

The time-series of the primary biomass burning factor observed by the EESI-TOF (Fig. 2b) correlates strongly with the summed time-series of the two BBOA factors from AMS ($R$=0.85). The diurnal trend of this factor shows a distinct peak during the evening rush hours between 18:00 LT - 22:00 LT and thereafter a steady decline throughout the night with an early morning rise starting between 05:00 LT - 06:00 LT and peaking at 08:00 LT before decreasing to low values during mid-day hours (12:00 LT - 16:00 LT) (Fig. 2c). The diurnal trend is also

qualitatively similar to the AMS HOA and BBOA factors, in that the time of early morning rise coincides with an increase in anthropogenic activities. Another observation is that during day-time (12:00-16:00) the primary biomass burning declines to less than 4 % of its average concentrations from the previous night. Three effects might drive these very low day-time concentrations of primary biomass burning. First, the decline in source intensities; second, the strong dilution effects from boundary layer expansion; and third increased evaporation of

semi-volatile constituents due to higher temperature and the aforementioned dilution. The primary biomass burning factor constitutes on average around 70 % of the total EESI-TOF OA signal, whereas the AMS BBOA factors contribute on average 44.7 % to the total organic mass measured by the AMS. This is due to the higher sensitivity of the EESI-TOF towards levoglucosan as compared to most other classes of compounds, as consistently observed in laboratory and field studies (Lopez-Hilfiker et al., 2019; Stefenelli et al., 2019; Tong et

al., 2020). (Note that the EESI-TOF insensitivity to HOA cannot explain this discrepancy, as AMS BBOA contributes only 50.9 % of the non-HOA organic mass, i.e. BBOA/ (OA-HOA)).

**Cooking-related OA:** A notable feature in the mass spectrum of this factor is that ~9.2 % of the total signal in this factor comes from ions with H:C ratios >1.7 and O:C < 0.25. The contribution of such ions to biogenic SOA is 5.0 %, with all other factors falling below 2.1 %. The carbon number distribution of these ions is shifted towards

higher carbon numbers, consistent with saturated and non-saturated fatty acids (see Fig. 3). Such molecules are prominent constituents of cooking oils (Orsavova et al., 2015). The high relative contributions from such species are consistent with previous cooking-related factors resolved in EESI-TOF studies (Tong et al., 2020; Qi et al., 2019).

The time series of cooking-related OA explains a large fraction of the signal from ions consistent with fatty acids,

e.g., 32.1 %, 38.8 %, 34.6 %, and 33.9 % of $C_{16}H_{32}O_2$, $C_{16}H_{30}O_2$, $C_{18}H_{34}O_2$, and $C_{18}H_{36}O_2$, respectively. The other 5 factors combined explain only 15-20 % of the variation of these fatty acid-like compounds, while the rest remains unexplained, ~5-10 % is explained by residuals, and ~90-95 % by noise. Fig. S3 shows the fractional contribution of all EESI-TOF factors to the diurnal trends of two selected fatty acid-like compounds, tentatively attributed to oleic acid ($C_{18}H_{34}O_2$) and stearic acid ($C_{18}H_{36}O_2$). As one can clearly see, the cooking-related OA

factor is the dominant contributor to these species, regardless of time of day or ion concentration. This observation of the high contribution of cooking-related factor to observed diurnal patterns of fatty acid-like compounds is also consistent with previously defined cooking-related OA factors from the EESI-TOF (Tong et al., 2020; Qi et al., 2019).





The diurnal trend of cooking-related OA shows qualitatively similar features to primary biomass burning, in that it decreases during day-time hours and peaks during the late evening (Fig. 2c). Cooking-related OA increases by a factor of ~6 from 16:00 LT -19:00 LT, remaining roughly stable till 00:00 LT, followed by a decline till 06:00 LT and then remaining approximately stable until 10:00 LT. During the day, a small peak is observed during

lunchtime (13:00 LT -15:00 LT) which may indicate active sources in the vicinity of the measurement site. The average day to night ratio is approximately a factor of 10 higher than that of primary biomass burning, which further supports the possibility of active sources during the day.

We note that no cooking-related factor was identified in the AMS PMF results. The unconstrained PMF analysis yielded an HOA factor with high levels of oxygenated fragments especially the oxygenated fragments at $m/z$ 55

and 57, which might indicate mixing of cooking-related factor into HOA. However, it is also possible that these oxygenated fragments were contributed by some other sources (e.g., BBOA or LO-OOA) hence a definite conclusion on the mixing of cooking related factor into unconstrained HOA could not be drawn. Neither constraining a COA profile from literature in the AMS PMF nor increasing the number of factors up to 15 yielded cooking-related factors. Possible reasons for this may be the similarity of the cooking-related OA spectrum with

the HOA and BBOA spectra in the AMS, high relative concentrations of the other primary factors, and strong effects of boundary layer dynamics on the diurnal patterns of all factors (leading to collinearity among unrelated factors), all of which combine to make it difficult to separate a relatively minor cooking-related factor without the specific tracer ions provided by the EESI-TOF.

### 3.2.2 Secondary factors

**Aromatic SOA:** The mass spectrum of this factor has ~63.0 % of its total signal contributed by compounds with 6 to 9 carbons ($C_6$-$C_9$). A large fraction of this comes from molecules with an H:C ratio < 1.5 (27.9 % of the total signal from $C_{6-9}H_yO_z$ ions and 7.1 % from $C_{6-9}H_yO_zN_{1-2}$ ions) (Fig. 3). As a comparison, biogenic SOA, aged biomass burning and mixed urban SOA have 11.7 %, 27.2 %, and 18.3 % contribution from $C_{6-9}H_yO_z$ ions and 4.0 %, 2.9 %, and 3.7 % contribution from $C_{6-9}H_yO_zN_{1-2}$ ions with H:C ratios < 1.5 respectively. The low H:C

ratios are associated with aromatic systems, as the precursor gases are highly unsaturated and contain fewer hydrogen atoms than more saturated straight-chain or ring-containing compounds. In a recent study on the source apportionment of VOCs at the same site, it was found that aromatic $C_{6-9}H_y$ VOCs constitute 45.4 % of total VOCs loading and are emitted into the atmosphere predominantly from anthropogenic activities, of which traffic constituted the highest fraction during day-time (Wang et al., 2020). Oxidation of these aromatic VOCs is most

likely the dominant process leading to the formation of this factor.

In order to substantiate that the major ions contributing to the ambient aromatic SOA factor are indeed formed by oxidation of aromatic VOCs, we conducted a chamber experiment in the Paul Scherrer Institute (PSI) smog chamber using a mixture of aromatic compounds consisting of benzene, toluene, ethylbenzene and trimethylbenzene (Kumar et al.; In prep). These compounds are well-established constituents of vehicular

emissions (Cao et al., 2016; Yao et al., 2015) especially for gasoline vehicles during the cold start phase (Platt et al., 2017). OH radicals were produced in the chamber and reacted with the VOCs, resulting in the formation of SOA whose chemical composition was subsequently compared with the aromatic SOA and other SOA factors obtained in this study.





Fig. S2 shows the mass spectrum of the ambient aromatic SOA factor color-coded by ions that were found in chamber SOA formed from oxidation of aromatics. Approximately 32.0 % of the EESI-TOF signal contained in the aromatic SOA factor overlapped with ions identified in the chamber experiment. This signal fraction is considerably higher than for other SOA factors i.e., biogenic SOA, aged biomass burning and mixed SOA had signal contribution from chamber SOA ions, of 16.0 %, 26.0 %, and 20.0 %, respectively.

Despite the strong dilution of the boundary layer during the day, there is little variation in the aromatic SOA concentration during the day. This points to a strong local day-time source, such that the production of aromatic SOA is fast enough to offset the strong dilution effects of the expanding boundary layer as discussed further in Sect. 3.5.

**Biogenic SOA:** The mass spectrum of biogenic SOA is shown in Fig. 2a. It contains high contributions from ions such as $C_9H_{16}O_5$, $C_8H_{14}O_6$, $C_9H_{14}O_4$, which have been previously identified in EESI-TOF factors representing biogenic oxidation products (Qi et al., 2020; Stefenelli et al., 2019). Compounds with H:C > 1.5 constitute nearly 50 % of the signal of this factor, with 2.9 %, 5.0 %, 6.8 %, and 2 % of signal resulting from $C_7H_{10-14}O_{4-8}$, $C_8H_{12-16}O_{4-8}$, $C_9H_{14-18}O_{4-8}$, and $C_{10}H_{16}O_z$ respectively. These compounds have been previously attributed to monoterpene oxidation products in Zurich, Switzerland (Stefenelli et al., 2019). There are, however some notable differences from the Zurich study. Specifically, the present study shows a much smaller contribution from the $C_{10}H_{16}O_z$ compounds (6.2 % in Zurich vs. 2 % in this study), while $C_xH_yO_zN_1$ compounds comprise 26.7% of the factor profile (vs. ~13% in Zurich). These differences might arise because of two different reasons. One is the probable contribution of not only monoterpene oxidation products (which dominate in Zurich) but also isoprene oxidation products, e.g., $C_5H_{10}O_x$ and $C_5H_9NO_x$ (Chen et al., 2020) to the biogenic factor retrieved in this study. This is because of the Delhi's location near the tropics, which results in a large contribution of isoprene to the total biogenic VOCs. Model estimates predict isoprene emissions fluxes to be a factor of ~20 higher than α-pinene in tropical regions of India, whereas in Europe the emission fluxes of isoprene and monoterpenes are similar (Guenther et al., 2012; Sindelarova et al., 2014). Zurich and Delhi also differ in terms of atmospheric conditions, in particular the much higher $NO_x$ levels in Delhi as compared to Zurich, consistent with the higher $C_xH_yO_zN_1$ fraction.

The diurnal trend of this factor resembles that of the EESI-PMF primary factors in that the concentration is highest overnight and a strong decrease is seen during day-time hours. A possible explanation of this behavior could be more regional sources of the biogenic VOCs scattered over a large area. This means that the biogenic SOA factor most likely has only a small day-time source in the vicinity of the site.

**Aged biomass burning:** The fractional contribution of levoglucosan ($C_6H_{10}O_5$) to the aged biomass burning factor mass spectrum is ~10 %, which is a factor of eight lower than for the primary biomass burning factor. The lower levoglucosan content in the aged biomass burning mass spectrum as compared to the primary biomass burning mass spectrum is consistent with observations of similar factors in Zurich during winter (Qi et al., 2019). Additionally, chamber studies have shown that the levoglucosan concentration decreases in aged biomass burning particles (Bertrand et al., 2018), while the concentrations of secondary species increase, consistent with observations in this study.

The mass spectrum of this factor also has ~2 % contributions each from two key ions, $C_6H_8O_6$ and $C_7H_8O_7$. These are most likely oxidation products of phenols and methoxy-phenols, which are abundant secondary compounds





formed during the aging of biomass burning emissions (Yee et al., 2013) and are important precursors of biomass burning SOA. The aged biomass burning factor contains dominant signals from various other small molecules with H:C ratios less than 1.3, which is consistent with the oxidation of small aromatic compounds emitted during biomass burning such as phenolic compounds. These compounds were observed to be major contributors of gas-phase solid-fuel combustion factors in a recent VOC source apportionment study conducted at the same site (Wang et al., 2020).

The diurnal trend of aged biomass burning is similar to the one of primary factors, characterized by increased concentrations during evening hours and a decline during day-time hours. The amplitude of the evening time peak however differs between aged biomass burning and primary biomass burning (Fig. S1). While the primary biomass burning increases by a factor of ~50-60 between 17:00 LT - 22:00 LT, the increase in oxidized biomass burning is within a factor of ~5 during the same time. Between 22:00 LT - 07:00 LT, the concentration of this factor steadily decays by a factor of 2.5. A distinct peak is also observed between 7:00 LT - 09:00 LT, which coincides with an increase in solar radiation indicating that aging of emissions takes place during early morning hours. Note that it is likely that a majority of local emissions are not oxidised during night-time in Delhi due to very high levels of NO (~200-300 ppb), which may scavenge both $O_3$ and $NO_3$ radicals during night-time and inhibit nocturnal degradation of VOCs (Haslett et al.; In prep)

The diurnal pattern of the LO-OOA factor from AMS correlates well with the aged biomass burning factor from the EESI-TOF and suggests that oxidation of biomass burning emissions may be the dominant contributor to the LO-OOA factor observed in the AMS.

**Mixed urban SOA:** The remaining factor is most likely a mixed SOA factor, which has influences from both anthropogenic and biogenic sources. The highest intensity ions in this factor are $C_5H_{10}O_4$, $C_9H_{14}O_5$, $C_6H_{10}O_5$, $C_9H_{16}O_5$, $C_9H_{18}O_5$, and $C_6H_{12}O_4$. $C_5H_{10}O_4$ is probably a product of isoprene oxidation, whereas the dominance of $C_9$ compounds suggests contributions from oxidation products of $C_9$ species The $C_9$ species have varied sources in urban areas which include evaporative losses of fuels (e.g., gasoline), solvent use, and unburnt exhaust emissions (Mehra et al., 2020; Zhang et al., 2013) and hence this factor is likely influenced by different sources linked to Delhi urban area. This factor also has ~2.5 % of mass spectral signal contributed by levoglucosan ($C_6H_{10}O_5$). This could be either due to non-perfect unmixing by PMF or it could indicate the contributions from biomass burning with other above-mentioned sources in this factor. This factor is therefore named mixed urban SOA.

The diurnal pattern of this factor shows roughly stable concentrations from 00:00 LT - 06:00 LT, which is similar to the aromatic SOA factor. All other factors show a decrease during these hours. It increases by a factor of ~3 between 11:00 LT - 14:00 LT and then steadily decays before being enhanced again by a factor of ~15 between 18:00 LT - 21:00 LT. The daytime increase suggests photochemical production of this factor. The diurnal trend of this factor is stable during late night hours and does not show a marked early morning rush hour peak, indicating little or no influence of morning rush hour emissions.

### 3.3 Estimation of mass contributions of EESI SOA factors

The EESI-TOF sensitivity towards individual compounds has been shown to vary by up to 1-2 orders of magnitude. Although EESI-TOF factor sensitivities likely vary by significantly less due to averaging effects, these





variations nonetheless make it challenging to ascertain relative contributions of EESI-TOF factors on a mass concentration basis. To estimate the EESI-TOF sensitivities (in counts per second/($\mu$g m$^{-3}$)) to different EESI SOA factors and thus obtain a mass-based source apportionment of the resolved SOA factors, a multiple linear regression (MLR) analysis was performed to explain the AMS SOA (i.e., MO-OOA + LO-OOA) time-series as a

function of the four EESI-TOF SOA factors. The Eq. 11 was solved for $\alpha_1$, $\alpha_2$, $\alpha_3$, and $\alpha_4$ where the reciprocal of the coefficients $\alpha_i$ represents the sensitivity of the EESI-TOF to each factor in cps per $\mu$g/m$^3$.

$$\text{AMS SOA} = \alpha_1 \times (\text{aromatic SOA}) + \alpha_2 \times (\text{biogenic SOA}) + \alpha_3 \times (\text{aged biomass burning}) + \alpha_4 \times (\text{mixed urban SOA}) + \in \qquad (11)$$

To solve Eq. 11, a weighted least-squares approach was used where the uncertainty weighted residuals (denoted $\in$) were minimized for each point in time. The $\alpha$ coefficients for all EESI-TOF SOA factors were constrained such

that the obtained sensitivities of these factors was between 0.1 to 1 times that of levoglucosan (~55 cps/$\mu$g m$^{-3}$), consistent with previous observations of bulk EESI-TOF sensitivities to SOA from different precursors (Lopez-Hilfiker et al., 2019). In addition to the MLR, the EESI sensitivities towards individual oxidation products were estimated using a gradient-boosting regression-prediction (GBRP) model (Wang et al., 2021) based on their elemental formulae (i.e. $C_xH_yO_z$). The EESI-TOF sensitivity to different SOA factors were derived by calculating

the  signal-weighted average based on factor profile of these individual ion sensitivities. The GBRP model results were used in relative terms, where the response factors obtained for each EESI-TOF factors were normalized relative to that of primary biomass burning.

The MLR analysis was first applied to the entire time series, which resulted in a correlation coefficient ($R$) value of 0.6 between  modeled SOA ($\alpha_1 \times$ aromatic SOA + $\alpha_2 \times$ biogenic SOA + $\alpha_3 \times$ aged biomass burning +

$\alpha_4 \times$ mixed urban SOA) and measured SOA (sum of MO-OOA and LO-OOA from AMS). There were, however two issues with this analysis. One was that it showed systematic positive and negative biases in certain parts of the time-series and second was that the fitted MLR coefficients for biogenic SOA, aged biomass burning, and mixed urban SOA were near-zero, which, based on previous studies, implied a non-physical result (Lopez-Hilfiker et al., 2019). The possible reason for these mentioned issues might be the presence of a unique event from 18:00

LT on 3 January 2019 to 12:00 LT on 4 January 2019 where high signals of aromatic SOA with low signals of biomass burning and other primary and secondary species were observed, which drives the coefficients of all other SOA factors except aromatic SOA to be near zero. Based on the issues mentioned above, the time series was divided into two parts: i.e., part-1 from 31 December 2018 - 3 January 2019 and part-2 from 4 January 2019 -13 January 2019. The data from 16:00 LT on 3 January 2019 to 12:00 LT on 4 January 2019 were omitted.

The sensitivities were then obtained by MLR analyses of different parts of the dataset: (1) the sensitivity was estimated by performing MLR on the entire EESI-TOF factor time-series; (2) the sensitivity was estimated by performing MLR on only part-1 of the time series; (3) the sensitivity was estimated by performing MLR on only part-2 of the time series; (4) the EESI-TOF sensitivity was assumed to be uniform for all factors, where the bulk EESI sensitivity was calculated as the slope of the total EESI-TOF signal vs. the total AMS organic mass. In

addition, the EESI-TOF sensitivity towards individual oxidation products was estimated using a gradient-boosting regression-prediction (GBRP) model (Wang et al., 2021) based on their elemental formulae (i.e. $C_xH_yO_z$) as described above.

The sensitivities obtained using MLR analysis and predicted by GBRP model were used to calculate modelled SOA and results were evaluated based on three parameters (Fig. S4): (1) Pearson correlation coefficient $R$ between





the modeled and measured SOA; (2) mean of the fractional residuals i.e., (measured SOA-modelled SOA/measured SOA); (3) mean of the scaled residuals i.e., (measured SOA-modelled SOA/uncertainty in measured SOA).

The fractional and scaled residuals were closest to zero and hence had the least bias when modeled SOA was calculated by using α coefficients obtained by MLR analysis on part-2 of the time series (Fig. S4). Hence the α coefficients obtained using MLR analysis on part-2 of the time series were applied to the entire time-series factors to calculate a mass-based estimation of the SOA factors. The coefficients obtained were 0.15, 0.11, 0.10, and 0.12 for aromatic SOA, biogenic SOA, aged biomass burning, and mixed urban SOA, respectively. These coefficients correspond to sensitivities of 6.6, 9.1, 10.0, and 8.3 cps/$\mu$g m$^{-3}$ respectively. As a comparison, the sensitivities predicted by Wang et al., (2021) were 6.1, 8.1, 8.6, and 11.1 cps/$\mu$g m$^{-3}$ for aromatic SOA, biogenic SOA, aged biomass burning and mixed urban SOA, respectively and lied between ±35 % of those obtained from MLR analysis providing evidence of robustness of MLR analysis. Fig. S5 shows the time-series of measured and modelled SOA obtained using the coefficients derived from the five different strategies discussed above.

### 3.4 Source apportionment of total OA

Here, we combine MLR-corrected EESI-TOF concentrations for aromatic SOA, biogenic SOA, aged biomass burning, and mixed urban SOA (the sum of which by definition approximates the total AMS-derived SOA) with the AMS source apportionment results for POA factors (i.e., HOA and BBOA) to provide an overall description of the OA sources influencing Delhi. The 24-h average, day-time average (10:00-16:00 LT), and night-time average (22:00-04:00 LT) factor contributions to total OA mass are shown in Fig. 4. While SOA contributes only 40.0 % of total OA on a daily average basis, there is a stark difference between day and night. SOA constitutes 76.8 % of total OA during the day-time (10:00 to 16:00 LT). The aromatic SOA is the largest contributor to day-time SOA and contributes 55.2 % of SOA during day-time (42.4 % of total OA), followed by biogenic SOA which contributes 18.4 % to day-time SOA (14.2 % to total OA). The contributions of aged biomass burning and mixed urban SOA to total SOA during day-time are 11.7 % and 8.5 % respectively (15.2 % and 11.0 % to total OA).

During night-time (21:00 to 04:00 LT), SOA constitutes 31.0 % of total OA mass, with biogenic SOA contributing 36.1 % of SOA (11.2 % of total OA) followed by 25.4 % contribution by aromatic SOA (7.9 % to total OA). Aged biomass burning and mixed urban SOA contribute 15.4 % and 22.9 % to total night-time SOA respectively (4.8 % and 7.1 % contribution to total OA respectively). During night-time, the high OA concentrations are driven by high primary emissions into a shallow boundary layer; during day-time, the OA is dominated by secondary aerosol, including local oxidation in the elevated boundary layer.

The differing relative contributions of primary vs. secondary OA as a function of time of day have implications for public health policy. Specifically, although POA dominates the overall OA concentration, the SOA factors are most prevalent during times when people are most likely to be outdoors and thus exposed to OA (i.e., daylight). It has been recently shown that oxygenated OA contributes a significantly higher fraction of particle bound-ROS (Zhou et al., 2019) as compared to primary OA. More specifically, anthropogenic SOA has been shown to be more relevant in terms of oxidative potential (OP) than biogenic SOA and POA (Daellenbach et al., 2020). In a recent study at the same site in Delhi, the ratio of hourly averaged ambient DTT (dithiothreitol) activity in PM$_{2.5}$ to the NR-PM$_1$ mass concentration (i.e., the intrinsic oxidative potential (OP$_{in}$)) was found to be highest during the afternoon period (Puthussery et al., 2020). This coincides with the increased contributions from





photochemically formed secondary organic aerosol (SOA) as observed in this study. Furthermore, the ratio of anthropogenic to biogenic SOA in Delhi especially during the day-time is high and the SOA fraction is dominated by aromatic SOA. This suggests that the day-time increase in $OP_{in}$ observed by Puthussery et al. (2020) is most likely driven by large contributions from aromatic SOA, similar to observations across Europe by (Daellenbach

et al., 2020). The aromatic SOA is most likely formed from the oxidation of light aromatics emitted by traffic. Reducing traffic emissions, e.g. by cleaning exhaust emissions with catalytic converters, can reduce emission factors of aromatic compounds and may lead to a decrease in total SOA concentration and oxidative potential of OA.

**3.5. Fraction of EESI-TOF factors attributable to local production or emissions during day-time**

As discussed previously, the temporal trends of the different factors are likely driven by photochemical production (SOA), emissions (POA), boundary layer dynamics, and gas-particle partitioning. Fig. 5 shows the estimated fraction of day-time concentrations attributable to photochemical production for the EESI-TOF SOA factors or direct emissions for EESI-TOF POA factors. AMS-derived PMF factors were not included in this analysis due to the lack of reliable methods to compute saturation vapour concentrations of these factors. The SOA factors

(aromatic SOA, aged biomass burning, and mixed urban SOA) have a high mean fraction of day-time photochemical production values of 0.88, 0.82, and 0.83, respectively. This is significantly higher than the day-time photochemical fraction of 0.55 for biogenic SOA (*t*-test: $p < 0.05$). This day-time production suggests that local photochemistry is an important driver for day-time air quality in Delhi, and thus relevant to human exposure and health outcomes. As shown in the VOC source apportionment study at the same site (Wang et al., 2020), the

largest contributor of primary VOCs during day-time at this site is a traffic-related factor. This is consistent with high concentrations of light aromatics, which are in turn consistent with the elevated concentration and strong local production term of the EESI-TOF aromatic SOA factor.

The smaller day-time production fraction retrieved for biogenic SOA is consistent with its description as a regionally-influenced factor. This is consistent with a projected source distribution that is diffused over a wide

area rather than limited to Delhi.

For the primary factors, a relatively small fraction i.e., 0.53 of primary biomass burning could be attributed to day-time emissions whereas a relatively high day-time emission fraction i.e., 0.83 was observed for cooking-related OA, consistent with expectations as primary biomass burning most likely corresponds to night-time heating activities, while the cooking-related emissions emerge from active sources during specific mealtimes.

**4. Conclusions**

Wintertime particulate air pollution in Delhi, India is a critical public health issue that affects millions of people. Previous studies have identified key POA sources contributing to this pollution, and suggested an important role of total SOA. Here we investigate the sources contributing to SOA via source apportionment of the first EESI-TOF deployment in India, in conjunction with AMS source apportionment.

The AMS source apportionment yielded POA factors related to traffic, primary biomass burning (2 factors), and SOA (2 factors), which in total comprised 60 and 40 % of the OA mass, respectively. The source apportionment of the EESI-TOF dataset yielded six factors. Two primary factors were identified as primary biomass burning and cooking-related OA, while the remaining four factors were attributed to secondary sources: aromatic SOA,





produced from the oxidation of light aromatics emitted by traffic; biogenic SOA, influenced by isoprene and monoterpene oxidation products and of regional influence; aged biomass burning; and mixed urban SOA, containing oxidation products consistent with a mix of sources and processes typical of the Delhi area. Multiple linear regression analysis (MLR) allowed us to calculate response factors for the EESI-TOF SOA factors and enabled apportionment the contribution of each EESI-TOF SOA factor to total SOA mass. During day-time, SOA dominated, comprising 76.8 % of the total OA mass with 42.4 % contribution from aromatic SOA. The night-time concentrations were dominated by POA, making up 69.0 % of total OA mass. Large variations in the relative contribution of SOA vs. POA to total OA were observed between day and night, with anthropogenic SOA sources being major contributors to day-time SOA explaining the previously observed day-time increase in OP of PM at the same site (Puthussery et al., 2020).

A simple partition and dilution modelling analysis was used to estimate the fraction of day-time concentrations that could be attributed to photochemical production for the SOA factors and emissions for the POA factors. Aromatic SOA was found to have the highest photochemical production among all SOA factors consistent with the high abundance of aromatic VOCs at the site as was previously seen (Wang et al., 2020). Biogenic SOA had significantly lower day-time photochemical production than other SOA factors indicating its regional nature and that its temporal behaviour is controlled by dilution and partitioning, and to a lesser extent by photochemical production. This study reveals that the HOA and BBOA are the main POA sources in Delhi and that aromatic SOA, biogenic SOA, aged biomass burning and mixed urban SOA constitute total SOA. The day time OA mass is dominated by SOA, which is mainly composed of aromatic SOA whereas the night-time OA is dominated by POA sources of which biomass burning is the dominant one.

**Data availability**

Data are available from the contact and corresponding authors upon request.

**Author contributions**

ASHP, JGS, and SNT designed the study and acquired the necessary funding. VK led the field campaign, did formal data analysis and wrote the manuscript. JGS and VK interpreted the data together. JGS, SG, SLH, YT, CPL, SM, RS, PV, JVP and DG provided the necessary support during the campaign. AS, JSD and NR provided AMS data. DSW provided GBRP model results. GS, VP, AB, RC, PR, SH, DB, LQ participated in campaign from PSI side. DMB, VV, CM, SNT, ASHP, and UB participated in the interpretation of data. All authors read and edited the manuscript.

**Competing Interests**

The authors declare that they have no conflict of interests.

**Acknowledgements**

This project was supported by the SDC Clean Air Project in India (grant no. 7F-10093.01.04), the Swiss National Science Foundation projects BSSGI0_155846 (IPR-SHOP),  200021_169787 (SAOPSOAG), IZLCZ2_169986 and by the European Union's Horizon 2020 Research and Innovation Program under the Marie Skłodowska-Curie grant agreement no. 701647. SNT gratefully acknowledges the financial support provided by the Department of



Biotechnology (DBT), Government of India, to conduct this research under grant no.644 BT/IN/UK/APHH/41/KB/2016-17 dated 19th July 2017 and the financial support provided by Central Pollution Control Board (CPCB), Government of India to conduct this research under grant no. AQM/Source apportionment_EPC Project/2017. SLH and CM gratefully acknowledge the financial support provided by Knut and Alice Wallenberg Foundation (WAF project CLOUDFORM, grant no. 2017.0165).

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

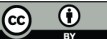



# Figures

**(a)**

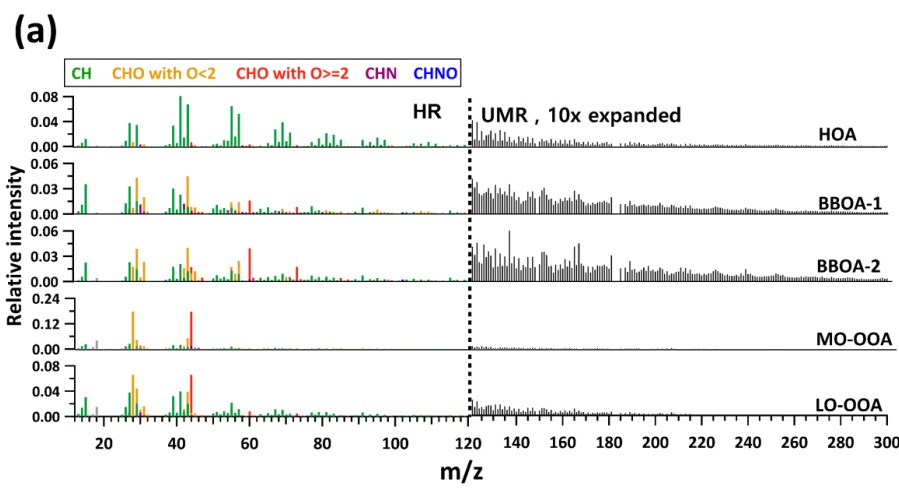

**(b)** **(c)**

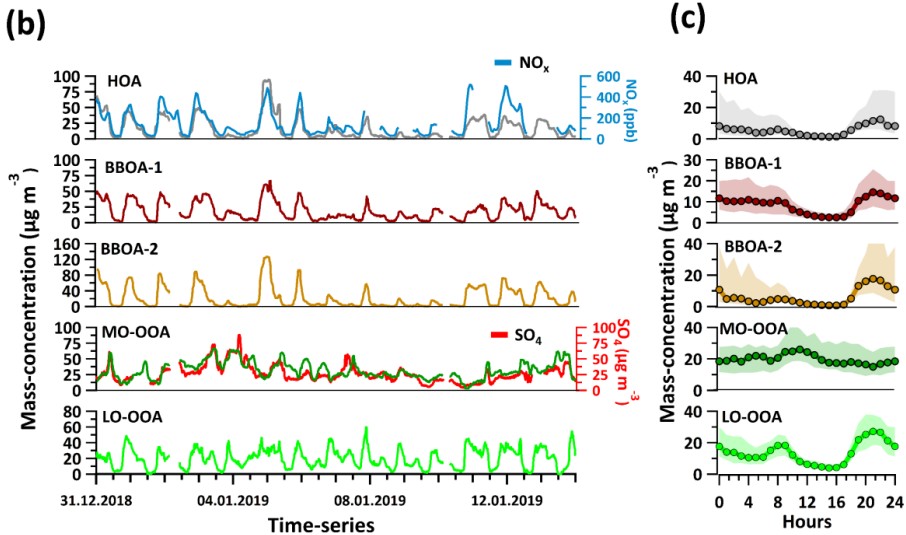

**(d)**

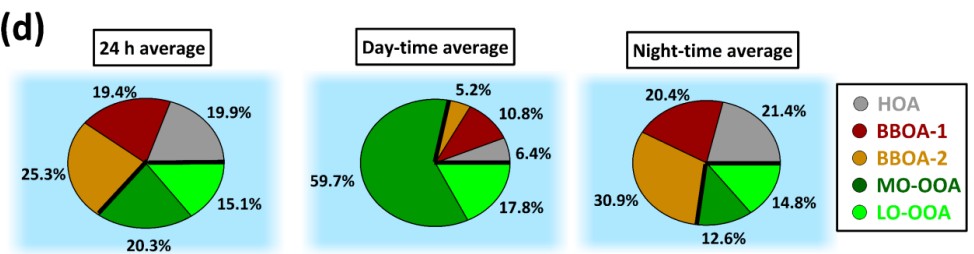





**Figure 1: (a)** Mass spectra from AMS PMF factors. The mass spectra is divided into two regions i.e., one from *m/z* 12 to 120 (individual ions from HR peak fitting) and from 121 to 300 (integer *m/z* integration). The mass spectra are coloured according to different families as mentioned in the legend. **(b)** Factor time series from AMS PMF results, together with selected reference species. **(c)** Diurnal trends with interquartile ranges (shaded areas)

5      of the AMS factors. These are drawn at an hourly time resolution **(d)** Pie-charts showing fractional contributions of AMS factors as 24-h average, as well as for day-time (10:00-16:00 LT) and night-time (21:00-04:00 LT).

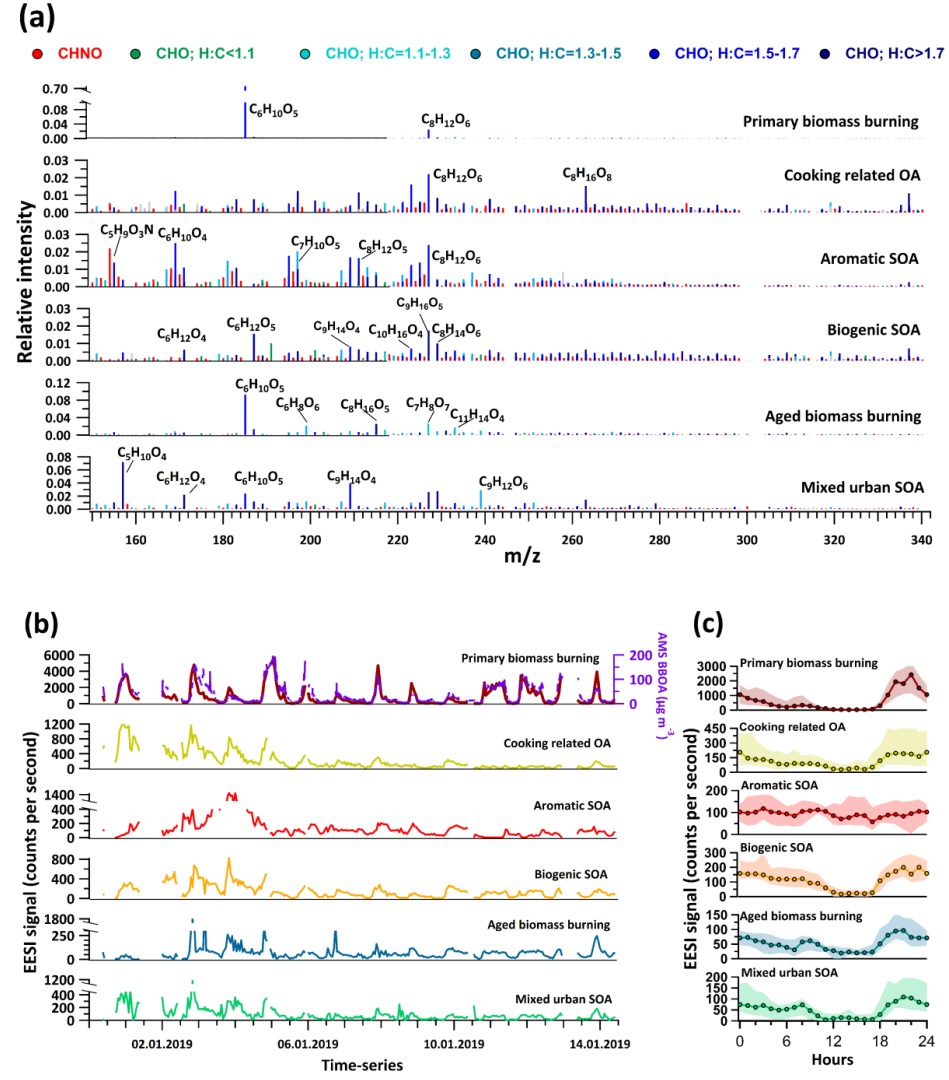

10      **Figure 2: (a)** Factor mass spectra of EESI-TOF PMF analysis. The green-to-blue color gradient represents $C_xH_yO_z$ compounds classified by their H:C ratio as mentioned in legend, while red denotes $C_xH_yO_zN_{1-2}$ compounds. **(b)**





Factor time series, together with selected external species for comparison. **(c)** Diurnal variations of the EESI-TOF

factors with an hourly resolution. Shaded areas show interquartile ranges.

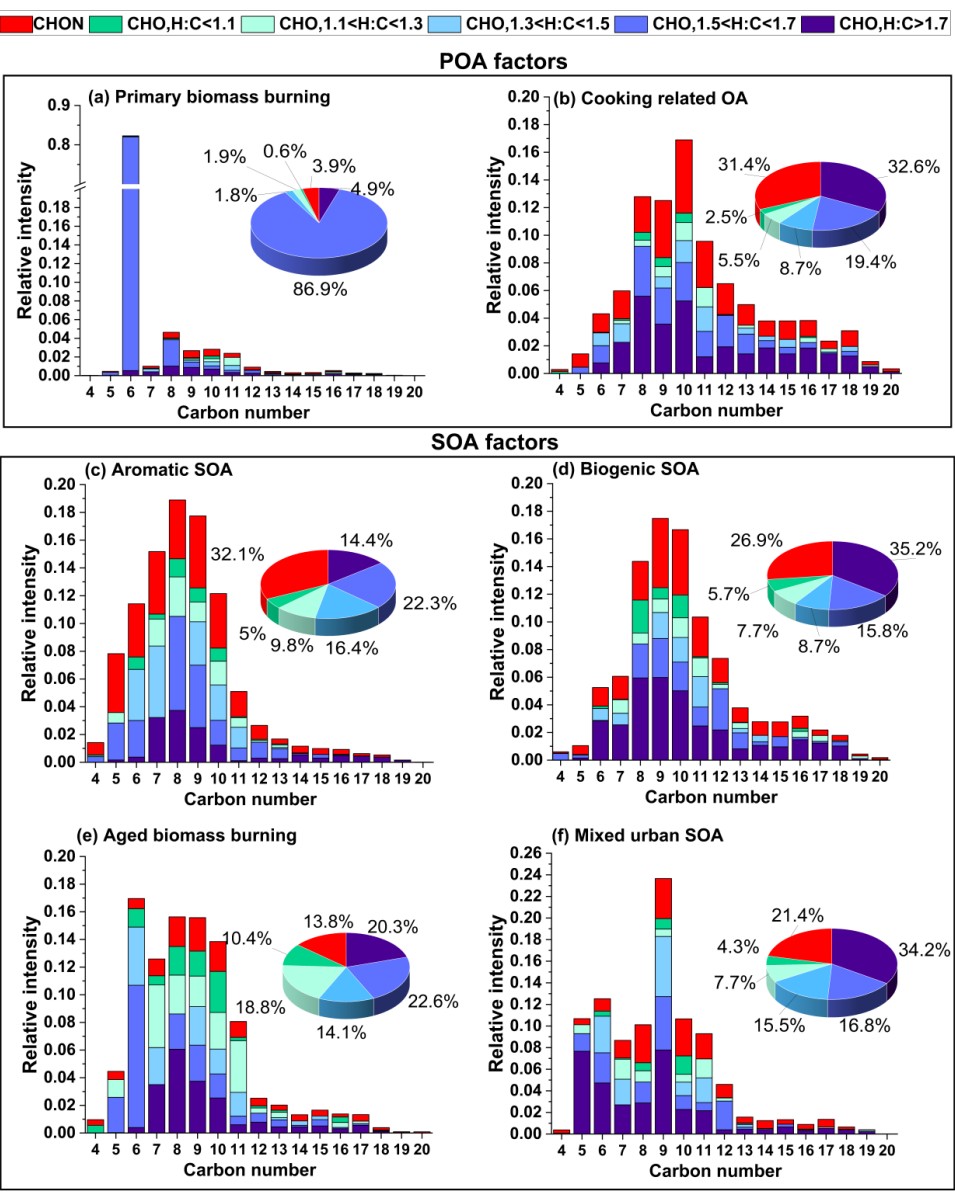

5 **Figure 3:** Carbon number distribution plots of EESI-TOF factors. Each carbon number contribution is stacked by

contributions from CHON species (red) and CHO species segregated by their H:C ratio categories. The green-to-





blue color gradient represents $C_xH_yO_z$ compounds classified by H:C ratios of: H:C < 1.1, 1.1 < H:C < 1.3, 1.3 < H:C < 1.5, 1.5 < H:C < 1.5 and H:C > 1.7 while red denotes $C_xH_yO_zN_{1-2}$.

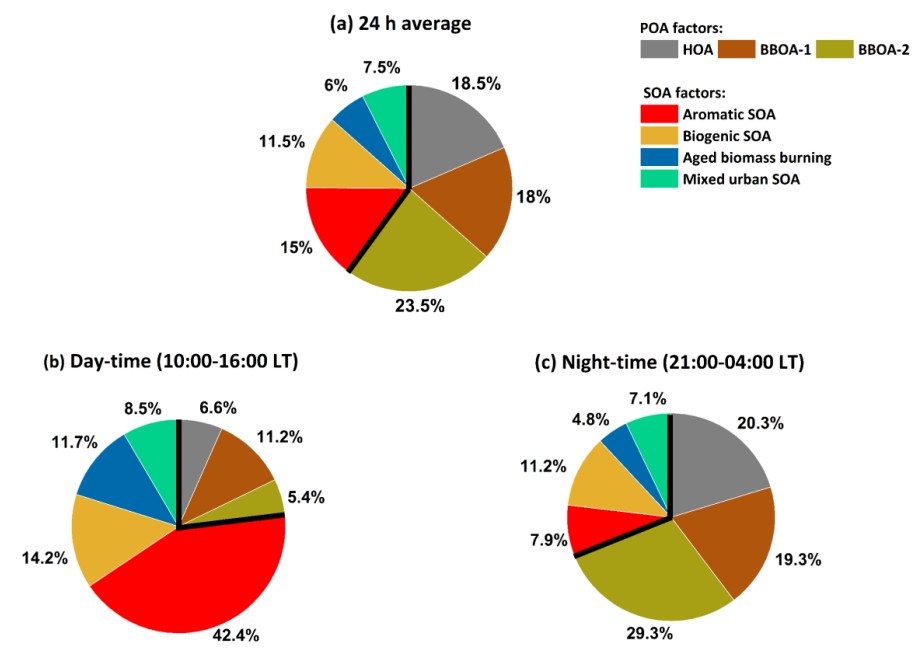

**Figure 4:** Overall OA source apportionment results, combining AMS PMF with MLR-corrected EESI-TOF SOA source apportionment results. EESI-TOF cooking-related OA is excluded. EESI-TOF primary biomass burning is assumed to be equivalent to the sum of the AMS BBOA factors, and therefore only the AMS factors are shown.

10 EESI-TOF SOA concentrations are calculated using the MLR-derived factor-dependent sensitivities. **(a)** shows the overall source apportionment, while **(b)** and **(c)** show the day-time and night-time results, respectively.





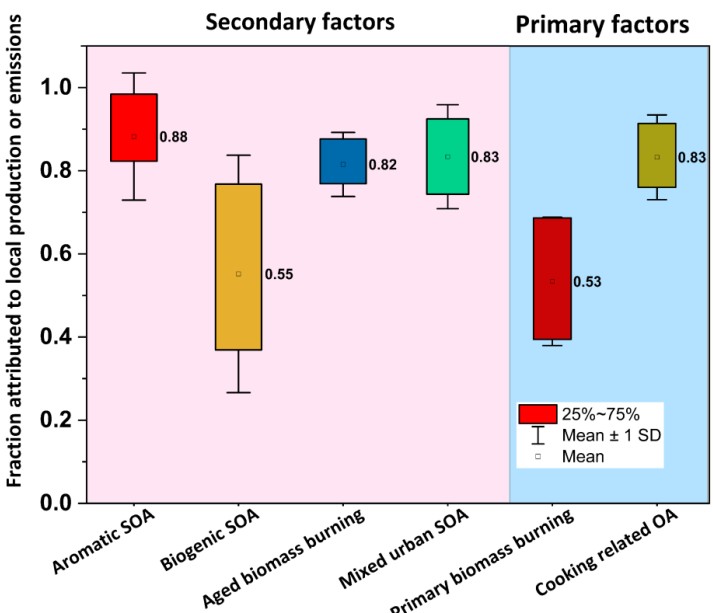

**Figure 5:** Box and whisker plots showing the fraction of day-time local production (SOA) or emission (POA) for EESI-TOF factors. The values averaged over all days for all factors are depicted by numbers adjacent to each box. For ease of viewing, background shading denotes SOA (blue) and POA (green) factors.

