# Peer review of "Real-time chemical speciation and source apportionment of organic aerosol components in Delhi, India, using extractive electrospray ionization mass spectrometry"

_Atmospheric Chemistry and Physics, 2021_

## Author Response (AR1)

**Response to RC1**

We thank the reviewer for the helpful comments. Below we provide a detailed point-by-point response to the issues raised by the reviewer. Reviewer comments are provided in *italics* and our responses follow in normal text. Changes to the manuscript are denoted in blue font.

**Comment #1**

*The authors inform a reader about disadvantages of widely used analytical instrumentation for aerosol characterisation and apply EESI and AMS for their study. I agree that EESI technique certainly has some advantages (which the authors briefly listed in the introductory section); however, as any other techniques, it has numerous limitations (e.g. sensitivity), which, I believe (to avoid any biases) need to be reflected in the introductory section. I believe for this reason, off-line organic analysis techniques are still widely applied for aerosol characterisation (Noziere et al., 2015). Another disadvantage of the later technique is that it employs electrospray ionisation which suffers from competitive ionisation and lead to signal enhancement or suppression. This is affected by a compound's functional group (-OH, -COOH) and presence of inorganic salts in the matrix that are important constituents of atmospheric aerosols (Noziere et al., 2015).*

For brevity, the original text did not discuss the advantages/disadvantages of offline organic analysis, addressing only continuous and semi-continuous online instrumentation. Offline techniques involve a different set of advantages/disadvantages relative to highly time-resolved online instrumentation such as the EESI-TOF. We have restructured the introduction to include a brief discussion of offline measurements in the discussion of analytical instrumentation.

The reviewer also raises two specific issues here: sensitivity of the EESI-TOF relative to offline techniques (where sensitivity can potentially refer to either the absolute detection limits or molecule-dependent differences in relative sensitivity), and matrix effects. We discuss these points individually below.

A comparison of absolute sensitivity is difficult to assess, due to (1) the large differences in time resolution between the EESI-TOF (typically seconds to a few minutes) and off-line techniques (typically hours to ~1 day), and (2) the potential for ongoing reactions and/or partitioning on the collection substrate to introduce systematic biases in off-line methods, introducing a disconnect between the bench-top and real-world off-line detection limits for affected compounds (Pospisilova et al., 2020; Zhao et al., 2018). As a result, we think it is not helpful to discuss absolute sensitivity in terms of advantages/disadvantages. Rather, this is an area where the different approaches aim at different targets and are thus complementary.

However, it is certainly true that different molecules exhibit different relative sensitivities in systems like the EESI-TOF and FIGAERO-I-CIMS that are based on soft ionization (Lopez-Hilfiker et al., 2019; Wang et al., 2021). Together with the lack of direct structural information, this is a clear limitation of 1-D MS techniques such as the EESI-TOF, AMS, and CHARON-PTR, as opposed to the chromatographic separation and tandem MS approaches possible in the off-line analysis. These points are noted in the revised manuscript.

Finally, the reviewer raises the issue of matrix effects. Although matrix effects do indeed affect conventional electrospray systems, they are well-known to be drastically reduced in EESI (Chen et al., 2006). In our EESI implementation, the combination of dilute analyte concentrations post-extraction in the charged droplets and the presence of high concentrations of $Na^+$ have been shown to render matrix effects negligible (Lee et al., 2021; Lopez-Hilfiker et al., 2019).

We now discuss this in the text as follows

Page 3, Line 38:

"To overcome these limitations on fragmentation and thermal decomposition, several offline, continuous and semi-continuous instruments have been developed. The offline techniques provide a high degree of chemically specific information with the possibility of molecular identification as well. They, however, have low time

resolution (typically hours to ~1 day) and include possible artifacts from reactions or partitioning on the surface (Pospisilova et al., 2020; Zhao et al., 2018)"

Page 4, line 19:

"The EESI-TOF enables highly time-resolved measurements of a wide range of atmospherically-relevant oxygenated compounds, including sugars, alcohols, acids, and organonitrates (Lopez-Hilfiker et al., 2019; Stefenelli et al., 2019) with detection limits on the order of 1-10 ng m$^{-3}$. The EESI-TOF detection limits are sufficient to measure these compounds with 5 s time resolution under typical ambient conditions with negligible thermal decomposition, ionization-induced fragmentation, or matrix effects. EESI-TOF provides the near molecular level information (i.e., molecular formula) with lack of direct structural information. This is a clear limitation of 1-D MS techniques such as the EESI-TOF, AMS, and CHARON-PTR, as opposed to the chromatographic separation and tandem MS approaches possible in the off-line analysis. In addition to that, different molecules exhibit different relative sensitivities in systems like the EESI-TOF (Lopez-Hilfiker et al., 2019; Wang et al., 2021)."

**Comment #2**

*The authors need to be more specific what they mean by atmospherically relevant compounds when stating detection limit "in order of 1-10 ng m$^{-3}$ for atmospherically relevant compounds" especially in light with the comments regarding competitive ionisation, selectivity and specificity of the applied technique. For example, PAHs (oxidised PAHs), sugar alcohols and carboxylic acids are atmospherically relevant compounds but have critically different ionisation efficiencies in ESI (and thus in EESI). Do you expect this technique to have the same "in order of 1-10 ng m$^{-3}$" detection limit for all of these atmospherically relevant compounds? If yes, please provide a reference or data to support this statement.*

We have revised the statement. We now note that the EESI-TOF measures a wide range of atmospherically-relevant oxygenated compounds, including sugars, alcohols, acids, and organonitrates (Lopez-Hilfiker et al., 2019; Stefenelli et al., 2019) The results of Tong et al. (submitted) suggest that on average the sensitivities between these compound classes are broadly consistent with the stated 1-10 ng m$^{-3}$ detection limits. However, variation among individual compounds is likely larger (Wang et al., 2021). We therefore now simply note that EESI-TOF detection limits are sufficient to measure these compounds with 5 s time resolution under typical ambient conditions. The modified text reads (page 4, line 19):

"The EESI-TOF enables highly time-resolved measurements of a wide range of atmospherically-relevant oxygenated compounds, including sugars, alcohols, acids, and organonitrates (Lopez-Hilfiker et al., 2019; Stefenelli et al., 2019) with detection limits on the order of 1-10 ng m$^{-3}$. The EESI-TOF detection limits are sufficient to measure these compounds with 5 s time resolution under typical ambient conditions"

**Comment #3**

*Including a reference for the following statement would be beneficial or support this by showing data: '' In the configuration of the mass spectrometer and ionization scheme used in this study, one can detect a wide range of molecules present in the organic aerosols, including sugars, alcohols, acids, and organo-nitrates.''*

We thank the reviewer for the suggestion. We have now added the following references with the text (page 4, line 19):

"The EESI-TOF enables highly time-resolved measurements of a wide range of atmospherically-relevant oxygenated compounds, including sugars, alcohols, acids, and organonitrates (Lopez-Hilfiker et al., 2019; Stefenelli et al., 2019) with detection limits on the order of 1-10 ng m$^{-3}$."

**Comment #4**

*The authors report a range of molecular formula detected by the technique and relate them to specific compounds, e.g., levoglucosan. I failed to find the mass accuracy and resolving power of the applied EESI technique to support their molecular assignments. Please add this to the paper and consider how this will impact on the presented results (molecular assignments).*

We thank the reviewer for the comment. The resolving power of the time-of-flight (TOF) mass analyser used in this study was ~8000 $M/dM$. This high resolving power of TOF analyser allowed the separation of isobars and the determination of molecular formula. A point to note here is that the molecular assignments given here are tentative and may also contain isomeric contributions. To clarify this, we have added the following lines to the text (page 6, line 8):

"The mass resolution ($M/dM$) achieved by the mass analyser in this study was ~8000. This resolution is enough to separate isobars (compounds with same nominal mass) and determine the molecular formula. Due to the lack of direct structural information, the molecular assignments given here are, however, tentative and may include contributions from multiple isomers."

**Comment #5**

*The authors associate $C_2H_4O_2^+$ (m/z 60) and $C_3H_5O_2^+$ (m/z 73) to levoglucosan. It is worth mentioning that other anhydrosugars ( levoglucosan isomers) can lead to this fragmentation. These include galactosan and mannosan, which are isomeric compounds of levoglucosan and cannot be distinguished/separated by the applied technique. Again, this caveat needs to be stated in the manuscript.*

We agree, and have changed the text in the manuscript as given below (page 14, line 14):

The mass spectrum of primary biomass burning is dominated by $C_6H_{10}O_5$, likely associated with anhydrosugars such as levoglucosan, mannosan and galactosan. $C_6H_{10}O_5$ constitutes 81.1 % of the total mass spectral signal in this factor (Fig. 2a).

**Comment #6**

*Line 10 (page 12) ''The mass spectra of BBOA-1 and BBOA-2 both have strong signals from $C_2H_4O_2^+$ (m/z 60) and $C_3H_5O_2$ + (m/z73) fragments, which are characteristic of anyhdrosugars like levoglucosan, a product of cellulose pyrolysis (Simoneit et al., 1999).'' needs revising. The work by Simoneit et al., 1999 reports data for TMS esters so the reference to levoglucosan fragments is invalid. If this reference was used to support the second part of the statement i.e., ''a product of cellulose pyrolysis'', then the sentence needs revising as well.*

The original text did not clearly link the references to the statements they were intended to support. The revised text reads (page 12, line 27):

The mass spectra of BBOA-1 and BBOA-2 both have strong signals from $C_2H_4O_2^+$ (*m/z* 60) and $C_3H_5O_2^+$ (*m/z* 73) fragments, which are characteristic fragments of anyhdrosugars like levoglucosan (Aiken et al., 2009), a product of cellulose pyrolysis (Hoffmann et al., 2010; Simoneit et al., 1999).

References

Aiken, A. C., Salcedo, D., Cubison, M. J., Huffman, J. A., DeCarlo, P. F., Ulbrich, I. M., Docherty, K. S., Sueper, D., Kimmel, J. R., Worsnop, D. R., Trimborn, A., Northway, M., Stone, E. A., Schauer, J. J., Volkamer, R. M., Fortner, E., de Foy, B., Wang, J., Laskin, A., … Jimenez, J. L. (2009). Mexico City aerosol analysis during MILAGRO using high resolution aerosol mass spectrometry at the urban supersite (T0) – Part 1: Fine particle composition and organic source apportionment. *Atmospheric Chemistry and Physics*, *9*(17), 6633–6653. https://doi.org/10.5194/acp-9-6633-2009

Chen, H., Venter, A., & Cooks, R. G. (2006). Extractive electrospray ionization for direct analysis of undiluted urine{,} milk and other complex mixtures without sample preparation. *Chem. Commun.*, *19*, 2042–2044. https://doi.org/10.1039/B602614A

Hoffmann, D., Tilgner, A., Iinuma, Y., & Herrmann, H. (2010). Atmospheric Stability of Levoglucosan: A Detailed Laboratory and Modeling Study. *Environmental Science & Technology*, *44*(2), 694–699. https://doi.org/10.1021/es902476f

Lee, C. P., Surdu, M., Bell, D. M., Lamkaddam, H., Wang, M., Ataei, F., Hofbauer, V., Lopez, B., Donahue, N. M., Dommen, J., Prevot, A. S. H., Slowik, J. G., Wang, D., Baltensperger, U., & El Haddad, I. (2021). Effects of aerosol size and coating thickness on the molecular detection using extractive electrospray ionization. *Atmospheric Measurement Techniques*, *14*(9), 5913–5923. https://doi.org/10.5194/amt-14-5913-2021

Lopez-Hilfiker, F. D., Pospisilova, V., Huang, W., Kalberer, M., Mohr, C., Stefenelli, G., Thornton, J. A., Baltensperger, U., Prevot, A. S. H., & Slowik, J. G. (2019). An extractive electrospray ionization time-of-flight mass spectrometer (EESI-TOF) for online measurement of atmospheric aerosol particles. *Atmospheric Measurement Techniques*, *12*(9), 4867–4886. https://doi.org/10.5194/amt-12-4867-2019

Pospisilova, V., Lopez-Hilfiker, F. D., Bell, D. M., El Haddad, I., Mohr, C., Huang, W., Heikkinen, L., Xiao, M., Dommen, J., Prevot, A. S. H., Baltensperger, U., & Slowik, J. G. (2020). On the fate of oxygenated organic molecules in atmospheric aerosol particles. *Science Advances*, *6*(11). https://doi.org/10.1126/sciadv.aax8922

Simoneit, B. R. T., Schauer, J. J., Nolte, C. G., Oros, D. R., Elias, V. O., Fraser, M. P., Rogge, W. F., & Cass, G. R. (1999). Levoglucosan, a tracer for cellulose in biomass burning and atmospheric particles. *Atmospheric Environment*, *33*(2), 173–182. https://doi.org/https://doi.org/10.1016/S1352-2310(98)00145-9

Stefenelli, G., Pospisilova, V., Lopez-Hilfiker, F. D., Daellenbach, K. R., Hüglin, C., Tong, Y., Baltensperger, U., Prévôt, A. S. H., & Slowik, J. G. (2019). Organic aerosol source apportionment in Zurich using an extractive electrospray ionization time-of-flight mass spectrometer (EESI-TOF-MS) -- Part 1: Biogenic influences and day--night chemistry in summer. *Atmospheric Chemistry and Physics*, *19*(23), 14825–14848. https://doi.org/10.5194/acp-19-14825-2019

Wang, D. S., Lee, C. P., Krechmer, J. E., Majluf, F., Tong, Y., Canagaratna, M. R., Schmale, J., Prévôt, A. S. H., Baltensperger, U., Dommen, J., El Haddad, I., Slowik, J. G., & Bell, D. M. (2021). Constraining the response factors of an extractive electrospray ionization mass spectrometer for near-molecular aerosol speciation. *Atmospheric Measurement Techniques*, 14, 6955–6972. https://doi.org/10.5194/amt-14-6955-2021

Tong Y., Qi, L., Stefenelli, G., Canonaco, D. S. W. F., Baltensperger, U., Prevot, A. S. H., & Jay G. Slowik. (2022). Quantification of primary and secondary organic aerosol sources by combined factor analysis of extractive electrospray ionisation and aerosol mass spectrometer measurements (EESI-TOF and AMS). *Submitted*

Zhao, R., Kenseth, C. M., Huang, Y., Dalleska, N. F., & Seinfeld, J. H. (2018). Iodometry-Assisted Liquid Chromatography Electrospray Ionization Mass Spectrometry for Analysis of Organic Peroxides: An Application to Atmospheric Secondary Organic Aerosol. *Environmental Science & Technology*, *52*(4), 2108–2117. https://doi.org/10.1021/acs.est.7b04863

**Response to RC2**

We thank the reviewer for the helpful comments. Below we provide a detailed point-by-point response to the issues raised by the reviewer. Reviewer comments are provided in *italics* and our responses follow in normal text. Changes to the manuscript are denoted in blue font.

**Comment #1**

*Page 8, L25: what is the reason behind removing signals related to $CO_2^+$, rather than down-weighting them?*

In standard AMS data analysis, $O^+$, $HO^+$, $H_2O^+$, and $CO^+$ are not directly measured but are instead calculated as constant fractions of the $CO_2^+$ signal. This must be accounted for in PMF to avoid overweighting the $CO_2^+$ variable. Two approaches are possible: (1) downweighting (by increasing the uncertainties) or (2) removing the variables and reinserting them afterwards. The two approaches are nearly equivalent, but may differ in some datasets due to the combination of the requirement of a minimum error for the AMS coupled with the dynamic downweighting routine applied to outliers in the "robust mode" of the PMF. This can lead to small differences in the effective signal/uncertainty ratios between $CO_2^+$ and its dependent ions, leading in turn to perturbations of these ratios in the output factors. As this is inconsistent with the basic data-processing assumptions for the $CO_2^+$-dependent ions, it should be avoided. Although it does not affect all data points, or even all datasets, there is the potential for this problem to occur when the $CO_2^+$ ions are retained for PMF. However, the remove-and-reinsert approach employed here is always safe, and thus preferred.

This is clarified in the text as follows (Page 9, Line 2):

"Note that this remove-and-reinsert strategy is preferable to downweighting of $CO_2^+$-dependent ions as it avoids the potential for small biases induced by the combination of AMS minimum errors and dynamic downweighting in "robust mode" operation of the PMF."

**Comment #2**

*When taking the HOA profile from the 8-factor solution set, do you mean you used SoFi to constrain the HOA factor? Similarly, on page 9, L 25, do you mean SoFi was used? It may be better to directly mention SoFi in these instances as well.*

We thank the reviewer for this comment. We have made following changes in the text:

(Page 9, line 13): "To get a cleaner HOA profile, we took the HOA factor profile from an unconstrained 8-factor solution and used it in SoFi to constrain the HOA factor in the final 5-factor solution"

(Page 10, line 4): "The reference profiles used in SoFi for these 5 factors were taken from the unconstrained 10-factor solution."

**Comment #3**

*Page 9, L26: how do we know the 5 factors from the 10-factor solution set were not mixed and reasonable to be assumed as pure reference profiles?*

The 5 factor profiles obtained from a 10-factor solution were obtained from the PMF model. Increasing the number of factors from 10 to 15 did not lead to further changes in the profiles of these 5 factors. This suggests that these factor profiles are robust and unmixed, at least in a mathematical sense. In the absence of evidence to the contrary or pre-existing reference profiles for the relevant sources/processes, these can be taken as stable profiles which are logical to assess for environmental interpretability. As discussed elsewhere in the text, these factors do prove

interpretable, and are while perhaps not completely free from mixing, provide our best estimate of the clean source profiles. Note also that the bootstrap/a-value analysis implies that the profiles from the 10-factor solution are only a starting point, with further adaptation permitted.

**Comment #4**

*Page 11, L7: Equation 7 does not assume that activity coefficient is 1. Do you mean in your calculation of $C^*$, you assumed it's one?*

Yes, we assumed it one. We have added this in the manuscript (Page 11, line 25):

"The activity coefficient was assumed to be 1"

**Comment #5**

*Figure S2 and page 16, L 4-5: Not having worked with EESI-ToF, I'm not sure how much fragmentation one gets. My understanding has been that it's a pretty soft ionization technique. If that's the case, I don't know how to interpret seeing signal at similar ions for both aromatic and biogenic SOA. To reconcile this, do you mean that the common signals are due to fragmentation?*

Since EESI-TOF cannot distinguish between structural isomers, we believe that common ions found in aromatic and biogenic SOA spectrum are actually isomeric compounds originating from different processes/ oxidation of different precursor VOCs. For example, Wang et al. (2021) showed substantial overlap in the molecular formulae between OH oxidation products of cresol (presumably from ring-opening reactions) and limonene. Similar results have been observed or inferred in other studies (Mehra et al., 2020; Stefenelli et al., 2019).

**Comment #6**

*Page 16, L 28: My first thought after reading that BSOA was high at night was that it's NO3-driven SOA. Later on, you mention that because of the high NO levels in Delhi you don't expect much of NO3 formation. Are NO levels so high that they titrate O3 completely such that NO2 conversion to NO3 is not possible? Even if that's the case, I think the potential for NO3+BVOC oxidation at night should be mentioned here.*

Our investigations suggest that the diurnal cycle for $N_2O_5$ and $NO_3$ in Delhi differs from many less-polluted urban areas, as discussed in detail in another study based on this campaign (Haslett et al., submitted). Because of very high levels of NO (~200-300 ppbv) in Delhi, the $NO_3$ and $N_2O_5$ formation is supressed in the boundary layer during the night-time. As a result, the concentrations of $N_2O_5$ during the night are substantially lower than in many other urban centres, while the highest concentrations are during the day. If night-time NO levels are considerably reduced, it can have substantial impacts on the production of night-time SOA. Following the reviewer's suggestion we have modified the text as follows (Page 17, line 6):

"Biogenic VOCs such as monoterpenes and isoprene are susceptible to oxidation by $NO_3$ radicals which can result in large amounts of biogenic SOA production. In Delhi, however, due to large concentrations of NO (~200-300 ppbv) during the night-time, the production of $NO_3$ radicals is suppressed and the diurnal cycle of $NO_3$ is actually inverted with the majority of available $NO_3$ radicals actually present during the daytime (Haslett et al., submitted)"

**Comment #7**

*Page 18, L 17: it is not clear to me what you mean by GBRP-based sensitivities in a relative sense. Please explain more.*

The GBRP-based sensitivities denote sensitivities estimated from the GBRP model, which are normalized to GBRP-based sensitivities calculated for the biomass burning factor. Specifically, the EESI-TOF response factor for biomass burning was calculated by taking the ratio of the summed EESI signal in primary biomass burning to the summed AMS BBOA factors. This was then used to scale the sensitivities of the SOA factors obtained using the GBRP model. We have added the following text to the manuscript (page 18, line 38):

"The GBRP model results were used in relative terms, where the response factors obtained for each EESI-TOF factors using the GBRP model were normalized relative to that of primary biomass burning. The EESI-TOF response factor for biomass burning was calculated by taking the ratio of the summed EESI signal in primary biomass burning to the summed AMS BBOA factors. This was then used to scale the sensitivities of the SOA factors obtained using the GBRP model."

**Comment #8**

*Page 18, L 31: case 1 still excludes the data from Jan 3-4, right? If so please indicate that clearly here too.*

Yes, the data from 18:00 LT on 3 January till 12:00 LT on 4 January were excluded. We have modified the text to state this clearly (page 19, line 14):

"Based on the issues mentioned above, the time series was divided into two parts: i.e., part-1 from 31 December 2018 - 3 January 2019 (till 18:00 LT) and part-2 from 4 January 2019 (from 12:00 LT) -13 January 2019. The data from 18:00 LT on 3 January 2019 to 12:00 LT on 4 January 2019 were omitted."

**Editorial:**

**Comment #9**

*Page 5, L2: do you mean PM10 and PM2.5, separately rather than respectively?*

We thank the reviewer for pointing this out. We indeed intended to mean "separately" here. We have now changed the text (Page 5, Line 9):

"to measure the mass of 35 different elements in $PM_{10}$ and $PM_{2.5}$, separately (Rai et al., 2020)."

**Comment #10**

*Page 11, L34: Please change to $C_xH_y^+$. Similarly, Page 13, L 11, please change to $SO_4^{2-}$ and on Page 18, L 14, change to $C_xH_yO_z^+$*

We thank the reviewer for the suggestions. We have now corrected the text.

Page 12, Line 16: $C_xH_y$ has been changed to $C_xH_y^+$

Page 13, Line 30: $SO_4$ has been changed to $SO_4^{2-}$

Page 19, Line 23: $C_xH_yO_z$ has been changed to $C_xH_yO_z^+$

**Comment #11**

*Figure S1: Consider changing SVOOA to LO-OOA to be consistent with other parts of the paper*

We thank the reviewer for the suggestion. We have changed SVOOA to LO-OOA in all parts of the manuscript

**Comment #12**

*Page 19, L 34: For completeness, please define ROS*

We thank the reviewer for the suggestion. We have defined ROS in the modified text (Page 20, Line 21):

"It has been recently shown that oxygenated OA contributes a significantly higher fraction of particle bound-reactive oxygen species (ROS) (Zhou et al., 2019) as compared to primary OA"

**Comment #13**

*Page 21, Line 5: Either change to "...enabled apportioning the ...." Or "...enabled apportionment of the ...."*

We thank the reviewer for the suggestion. We have modified the text (Page 21, Line 30):

"enabled apportioning the"

**Comment #14**

*Figure 5. Figure caption indicates colors of the background that are different than the real colors for secondary and primary factors.*

We thank the reviewer for pointing this out. We have changed the caption of Fig. 5 to indicate accurate colors of the background shading

"For the ease of viewing, background shading denotes SOA (pink) and POA (blue) factors"

**References**

Haslett, S. L., Bell, D. M., Kumar, V., Slowik, J. G., Mishra, S., Rastogi, N., Singh, A., Ganguly, D., Thornton, J., Dällenbach, K., Yan, C., Kulmala, M., Liu, Y., Zheng, F., Nie, W., Li, Y., Tripathi, S. N., & Mohr, C. (2022). Night-time NO emissions suppress large amounts of chlorine radical formation in Delhi. *Submitted*.

Mehra, A., Wang, Y., Krechmer, J. E., Lambe, A., Majluf, F., Morris, M. A., Priestley, M., Bannan, T. J., Bryant, D. J., Pereira, K. L., Hamilton, J. F., Rickard, A. R., Newland, M. J., Stark, H., Croteau, P., Jayne, J. T., Worsnop, D. R., Canagaratna, M. R., Wang, L., & Coe, H. (2020). Evaluation of the chemical composition of gas- and particle-phase products of aromatic oxidation. *Atmospheric Chemistry and Physics*, *20*(16), 9783–9803. https://doi.org/10.5194/acp-20-9783-2020

Rai, P., Furger, M., El Haddad, I., Kumar, V., Wang, L., Singh, A., Dixit, K., Bhattu, D., Petit, J.-E., Ganguly, D., Rastogi, N., Baltensperger, U., Tripathi, S. N., Slowik, J. G., & Prévôt, A. S. H. (2020). Real-time measurement and source apportionment of elements in Delhi's atmosphere. *Science of The Total Environment*, *742*, 140332. https://doi.org/https://doi.org/10.1016/j.scitotenv.2020.140332

Stefenelli, G., Pospisilova, V., Lopez-Hilfiker, F. D., Daellenbach, K. R., Hüglin, C., Tong, Y., Baltensperger, U., Prévôt, A. S. H., & Slowik, J. G. (2019). Organic aerosol source apportionment in Zurich using an extractive electrospray ionization time-of-flight mass spectrometer (EESI-TOF-MS) -- Part 1: Biogenic influences and day--night chemistry in summer. *Atmospheric Chemistry and Physics*, *19*(23), 14825–14848. https://doi.org/10.5194/acp-19-14825-2019

Wang, D. S., Lee, C. P., Krechmer, J. E., Majluf, F., Tong, Y., Canagaratna, M. R., Schmale, J., Prévôt, A. S.

H., Baltensperger, U., Dommen, J., El Haddad, I., Slowik, J. G., & Bell, D. M. (2021). Constraining the response factors of an extractive electrospray ionization mass spectrometer for near-molecular aerosol speciation. *Atmospheric Measurement Techniques Discussions*, *2021*, 1–24. https://doi.org/10.5194/amt-2021-125

Zhou, J., Elser, M., Huang, R.-J., Krapf, M., Fröhlich, R., Bhattu, D., Stefenelli, G., Zotter, P., Bruns, E. A., Pieber, S. M., Ni, H., Wang, Q., Wang, Y., Zhou, Y., Chen, C., Xiao, M., Slowik, J. G., Brown, S., Cassagnes, L.-E., … Dommen, J. (2019). Predominance of secondary organic aerosol to particle-bound reactive oxygen species activity in fine ambient aerosol. *Atmospheric Chemistry and Physics*, *19*(23), 14703–14720. https://doi.org/10.5194/acp-19-14703-2019